# FRAME AVERAGING FOR INVARIANT AND EQUIVARIANT NETWORK DESIGN

**Omri Puny**[*1]   **Matan Atzmon**[*1]   **Heli Ben-Hamu**[*1]   **Ishan Misra**[2]
**Aditya Grover**[2]   **Edward J. Smith**[2]   **Yaron Lipman**[2] [1]
[1]Weizmann Institute of Science   [2]Facebook AI Research

## ABSTRACT

Many machine learning tasks involve learning functions that are known to be invariant or equivariant to certain symmetries of the input data. However, it is often challenging to design neural network architectures that respect these symmetries while being expressive and computationally efficient. For example, Euclidean motion invariant/equivariant graph or point cloud neural networks.

We introduce Frame Averaging (FA), a general purpose and systematic framework for adapting known (backbone) architectures to become invariant or equivariant to new symmetry types. Our framework builds on the well known group averaging operator that guarantees invariance or equivariance but is intractable. In contrast, we observe that for many important classes of symmetries, this operator can be replaced with an averaging operator over a small subset of the group elements, called a frame. We show that averaging over a frame guarantees exact invariance or equivariance while often being much simpler to compute than averaging over the entire group. Furthermore, we prove that FA-based models have maximal expressive power in a broad setting and in general preserve the expressive power of their backbone architectures. Using frame averaging, we propose a new class of universal Graph Neural Networks (GNNs), universal Euclidean motion invariant point cloud networks, and Euclidean motion invariant Message Passing (MP) GNNs. We demonstrate the practical effectiveness of FA on several applications including point cloud normal estimation, beyond 2-WL graph separation, and $n$-body dynamics prediction, achieving state-of-the-art results in all of these benchmarks.

## 1   INTRODUCTION

Many tasks in machine learning (ML) require learning functions that are invariant or equivariant with respect to symmetric transformations of the data. For example, graph classification is invariant to a permutation of its nodes, while node prediction tasks are equivariant to node permutations. Consequently, it is important to design expressive neural network architectures that are by construction invariant or equivariant for scalable and efficient learning. This recipe has proven to be successful for many ML tasks including image classification and segmentation (LeCun et al., 1998; Long et al., 2015), set and point-cloud learning (Zaheer et al., 2017; Qi et al., 2017a), and graph learning (Kipf & Welling, 2016; Gilmer et al., 2017; Battaglia et al., 2018).

Nevertheless, for some important instances of symmetries, the design of invariant and/or equivariant networks is either illusive (Thomas et al., 2018; Dym & Maron, 2020), computationally expensive or lacking in expressivity (Xu et al., 2018a; Morris et al., 2019; Maron et al., 2019; Murphy et al., 2019). In this paper, we propose a new general-purpose framework, called Frame Averaging (FA), that can systematically facilitate expressive invariant and equivariant networks with respect to a broad class of groups. At the heart of our framework, we build on a basic fact that arbitrary functions $\phi : V \to \mathbb{R}$, $\Phi : V \to W$, where $V, W$ are some vector spaces, can be made invariant or equivariant by *symmetrization*, that is averaging over the group (Yarotsky, 2021; Murphy et al., 2018), i.e.,

$$\psi(X) = \frac{1}{|G|} \sum_{g \in G} \phi(g^{-1} \cdot X) \quad \text{or} \quad \Psi(X) = \frac{1}{|G|} \sum_{g \in G} g \cdot \Phi(g^{-1} \cdot X). \tag{1}$$

---

[*]Equal contribution

where $G = \{g\}$ denotes the group, $\psi : V \to \mathbb{R}$ is invariant and $\Psi : V \to W$ is equivariant with respect to $G$. Furthermore, since invariant and equivariant functions are fixed under group averaging, i.e., $\psi = \phi$ for invariant $\phi$ and $\Psi = \Phi$ for equivariant $\Phi$, the above scheme often leads to universal (i.e., maximally expressive) models (Yarotsky, 2021). However, the challenge with equation 1 is that when the cardinality of $G$ is large (e.g., combinatorial groups such as permutations) or infinite (e.g., continuous groups such as rotations), then exact averaging is intractable. In such cases, we are forced to approximate the sum via heuristics or Monte Carlo (MC), thereby sacrificing the exact invariance/equivariance property for computational efficiency, e.g., Murphy et al. (2018; 2019) define heuristic averaging strategies for approximate permutation invariance in GNNs; similarly, Hu et al. (2021) and Shuaibi et al. (2021) use MC averaging for approximate rotation equivariance in GNNs. A concurrent approach is to find cases where computing the symmetrization operator can be done more efficiently (Sannai et al., 2021).

The key observation of the current paper is that the group average in equation 1 can be replaced with an average over a carefully selected subset $\mathcal{F}(X) \subset G$ while retaining both exact invariance/equivariance and expressive power. Therefore, if $\mathcal{F}$ can be chosen so that the cardinality $|\mathcal{F}(X)|$ is mostly small, averaging over $\mathcal{F}(X)$ results in both expressive and efficient invariant/equivariant model. We call the set-valued function $\mathcal{F} : V \to 2^G$, a *frame*, and show that it can successfully replace full group averaging if it satisfies a *set equivariance property*. We name this framework Frame Averaging (FA) and it serves as the basis for the design of invariant/equivariant networks in this paper.

We instantiate the FA framework by considering different choices of symmetry groups $G$, their actions on data spaces $V, W$ (manifested by choices of group representations), and the backbone architectures (or part thereof) $\phi, \Phi$ we want to make invariant/equivariant to $G$. We consider: (i) Multi-Layer Perceptrons (MLP), and Graph Neural Networks (GNNs) with node identification (Murphy et al., 2019; Loukas, 2020) adapted to permutation invariant Graph Neural Networks (GNNs); (ii) Message-Passing GNN (Gilmer et al., 2017) adapted to be invariant/equivariant to Euclidean motions, $E(d)$; (iii) Set network, DeepSets and PointNet (Zaheer et al., 2017; Qi et al., 2017a) adapted to be equivariant or *locally* equivariant to $E(d)$; (iv) Point cloud network, DGCNN (Wang et al., 2018), adapted to be equivariant to $E(d)$.

Theoretically, we prove that the FA framework maintains the expressive power of its original backbone architecture which leads to some interesting corollaries: First, (i) results in invariant universal graph learning models; (ii) is an $E(d)$ invariant/equivariant GNN that maintain the power of message passing (Xu et al., 2018a; Morris et al., 2019); and (iii), (iv) furnish a universal permutation and $E(d)$ invariant/equivariant models. We note that both the construction and the proofs are arguably considerably simpler than the existing alternative constructions and proofs for this type of symmetry (Thomas et al., 2018; Fuchs et al., 2020; Dym & Maron, 2020). We experimented with FA on different tasks involving symmetries including: point-cloud normal estimation, beyond 2-Weisfeiler-Lehman graph separation, and $n$-body dynamics predictions, reaching state of the art performance in all.

## 2 FRAME AVERAGING

In this section we introduce the FA approach using a generic formulation; in the next section we instantiate FA to different problems of interest.

### 2.1 FRAME AVERAGING FOR FUNCTION SYMMETRIZATION

Let $\phi : V \to \mathbb{R}$ and $\Phi : V \to W$ be some arbitrary functions, where $V, W$ are normed linear spaces with norms $\|\cdot\|_V, \|\cdot\|_W$, respectively. For example, $\phi, \Phi$ can be thought of as neural networks. We consider a group $G = \{g\}$ that describes some symmetry we want to incorporate into $\phi, \Phi$. The way the symmetries $g \in G$ are applied to vectors in $V, W$ is described by the group's *representations* $\rho_1 : G \to \text{GL}(V)$, and $\rho_2 : G \to \text{GL}(W)$, where $\text{GL}(V)$ is the space of invertible linear maps $V \to V$ (automorphisms). A representation $\rho_i$ preserves the group structure by satisfying $\rho_i(gh) = \rho_i(g)\rho_i(h)$ for all $g, h \in G$ (see e.g., Fulton & Harris (2013)). As customary, we will sometimes refer to the linear spaces $V, W$ as representations.

Our goal is to make $\phi$ into an *invariant* function, namely satisfy $\phi(\rho_1(g)X) = \phi(X)$, for all $g \in G$ and $X \in V$; and $\Phi$ into an *equivariant* function, namely $\Phi(\rho_1(g)X) = \rho_2(g)\Phi(X)$, for all $g \in G$ and $X \in V$. We will do that by averaging over group elements, but instead of averaging over the entire group every time (as in equation 1) we will average on a subset of the group elements called a *frame*.

**Definition 1.** *A frame is defined as a set valued function $\mathcal{F} : V \to 2^G \smallsetminus \varnothing$.*

1. *A frame is $G$-equivariant if $\mathcal{F}(\rho_1(g)X) = g\mathcal{F}(X), \quad \forall X \in V, \ g \in G$, where as usual, $g\mathcal{F}(X) = \{gh \mid h \in \mathcal{F}(X)\}$, and the equality should be understood as equality of sets.*

2. *A frame is* bounded *over a domain $K \subset V$ if there exists a constant $c > 0$ so that $\|\rho_2(g)\|_{\mathrm{op}} \le c$, for all $g \in \mathcal{F}(X)$ and all $X \in K$, where $\|\cdot\|_{\mathrm{op}}$ denotes the induced operator norm over $W$.*

Figure 1 provides an illustration. How are equivariant frames useful? Consider a scenario where an equivariant frame is easy to compute, and furthermore its cardinality, $|\mathcal{F}(X)|$, is not too large. Then averaging over the frame, denoted $\langle \cdot \rangle_{\mathcal{F}}$ and defined by

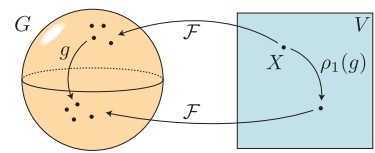

$$\langle \phi \rangle_{\mathcal{F}}(X) = \frac{1}{|\mathcal{F}(X)|} \sum_{g \in \mathcal{F}(X)} \phi(\rho_1(g)^{-1}X) \qquad (2)$$

$$\langle \Phi \rangle_{\mathcal{F}}(X) = \frac{1}{|\mathcal{F}(X)|} \sum_{g \in \mathcal{F}(X)} \rho_2(g)\Phi(\rho_1(g)^{-1}X) \qquad (3)$$

Figure 1: Frame equivariance (sphere shape represents the group $G$; square represents $V$).

provides the required function symmetrization. In Appendix A.1 we prove:

**Theorem 1** (Frame Averaging). *Let $\mathcal{F}$ be a $G$ equivariant frame, and $\phi : V \to \mathbb{R}$, $\Phi : V \to W$ some functions. Then, $\langle \phi \rangle_{\mathcal{F}}$ is $G$ invariant, while $\langle \Phi \rangle_{\mathcal{F}}$ is $G$ equivariant.*

Several comments are in order: First, the invariant case (equation 2) is a particular case of the equivariant case (equation 3) under the choice of $W = \mathbb{R}$ and the trivial representation $\rho_2(g) \equiv 1$. Second, in this paper we only consider $X$ and frame choices $\mathcal{F}$ for which $\mathcal{F}(X)$ are finite sets. Nevertheless, treating the infinite case is an important future research direction. Third, a trivial choice of an equivariant frame is $\mathcal{F}(X) \equiv G$, that is, taking the frame to be the entire group for all $X \in V$ (for infinite but compact $G$ the sum in the FA in this case can be replaced with Harr integral). This choice can be readily checked to be equivariant, and turns the FA equations 2, 3 into standard group averaging operators, equation 1. The problem with this choice, however, is that it often results in an intractable or challenging computation, e.g., when the group is large or infinite. In contrast, as we show below, in some useful cases one can compute a manageable size frame and can use it to build invariant or equivariant operators in a principled way. Let us provide a simple example for Frame Averaging: consider $V = \mathbb{R}^n$, $W = \mathbb{R}$, and $G = \mathbb{R}$ with addition as the group action. We choose the group actions[1] in this case to be $\rho_1(a)\boldsymbol{x} = \boldsymbol{x} + a\mathbf{1}$, and $\rho_2(a)b = b + a$, where $a, b \in \mathbb{R}$, $\boldsymbol{x} \in \mathbb{R}^n$, and $\mathbf{1} \in \mathbb{R}^n$ is the vector of all ones. We can define the frame in this case using the averaging operator $\mathcal{F}(\boldsymbol{x}) = \left\{ \frac{1}{n}\mathbf{1}^T\boldsymbol{x} \right\} \subset G = \mathbb{R}$. Note that in this case the frame contains only one element from the group, in other cases finding such a small frame is hard or even impossible. One can check that this frame is equivariant per Definition 1. The FA of $\phi : \mathbb{R}^n \to \mathbb{R}$ would be $\langle \phi \rangle_{\mathcal{F}}(\boldsymbol{x}) = \phi(\boldsymbol{x} - \frac{1}{n}(\mathbf{1}^T\boldsymbol{x})\mathbf{1})$ in the invariant case, and $\langle \phi \rangle_{\mathcal{F}}(\boldsymbol{x}) = \phi(\boldsymbol{x} - \frac{1}{n}(\mathbf{1}^T\boldsymbol{x})\mathbf{1}) + \frac{1}{n}\mathbf{1}^T\boldsymbol{x}$ in the equivariant case.

**Incorporating $G$ as a second symmetry.** An important use case of frame averaging is with the backbones $\phi, \Phi$ already invariant/equivariant w.r.t. some symmetry group $H$ and our goal is to make it invariant/equivariant to $H \times G$. For example, say we want to add $G = E(3)$ equivariance to permutation invariant set or graph functions, i.e., $H = S_n$. We will provide sufficient conditions for the FA to provide this desired invariance/equivariance. First, let us assume $H$ is acting on $V$ and $W$ by the representations $\tau_1 : H \to \mathrm{GL}(V)$ and $\tau_2 : H \to \mathrm{GL}(W)$, respectively. Assume $\phi$ is $H$ invariant and $\Phi$ is $H$ equivariant. We say that representations $\rho_1$ and $\tau_1$ *commute* if $\rho_1(g)\tau_1(h)X = \tau_1(h)\rho_1(g)X$ for all $g \in G$, $h \in H$, and $X \in V$. If $\rho_1$ and $\tau_1$ commute then the map $\gamma_1 : H \times G \to \mathrm{GL}(V)$ defined by $\gamma_1(h, g) = \tau_1(h)\rho_1(g)$ is a representation of the group $H \times G$. Second, we would need that the frame $\mathcal{F}(X)$ is *invariant* to $H$, that is $\mathcal{F}(\tau_1(h)X) = \mathcal{F}(X)$. We show a generalization of Theorem 1:

**Theorem 2** (Frame Average second symmetry). *Assume $\mathcal{F}$ is $H$-invariant and $G$-equivariant. Then,*

1. *If $\phi : V \to \mathbb{R}$ is $H$ invariant and $\rho_1, \tau_1$ commute then $\langle \phi \rangle_{\mathcal{F}}$ is $G \times H$ invariant.*

2. *If $\Phi : V \to W$ is $H$ equivariant and $\rho_i, \tau_i$, $i = 1,2$, commmute then $\langle \Phi \rangle_{\mathcal{F}}$ is $G \times H$ equivariant.*

---

[1]Note that since these are affine maps they are technically not representations but have an equivalent representation using homogeneous coordinates. Therefore, FA is also valid with affine actions as used here.

**Right actions.** Above we used left actions for the definition of equivariance. There are other flavors of equivariance, e.g., if one of the actions is right. For example, if $g$ multiplies $\mathcal{F}(X)$ from the right, then equivariance will take the form:

$$\mathcal{F}(\rho_1(g)X) = \mathcal{F}(X)g^{-1}, \quad \forall X \in V, \ g \in G \tag{4}$$

and accordingly

$$\langle\phi\rangle_{\mathcal{F}}(X) = \frac{1}{|\mathcal{F}(X)|} \sum_{g \in \mathcal{F}(X)} \phi(\rho_1(g)X), \quad \langle\Phi\rangle_{\mathcal{F}}(X) = \frac{1}{|\mathcal{F}(X)|} \sum_{g \in \mathcal{F}(X)} \rho_2(g)^{-1}\Phi(\rho_1(g)X) \tag{5}$$

are $G$ invariant and equivariant, respectively.

**Efficient calculation of invariant frame averaging.** There could be instances of the FA framework (indeed we discuss such a case later) where $|\mathcal{F}(X)|$ is still too large to evaluate equations 2,3. In the invariant case, there is a more efficient form of FA, that can potentially be applied. To show it, let us start by defining the subgroup of symmetries of $X$, i.e., its stabilizer. The stabilizer of an element $X \in V$ is a subgroup of $G$ defined by $G_X = \{g \in G \mid \rho_1(g)X = X\}$. $G_X$ naturally induces an equivalence relation $\sim$ on $\mathcal{F}(X)$, with $g \sim h \iff hg^{-1} \in G_X$. The equivalence classes (orbits) are $[g] = \{h \in \mathcal{F}(X) | g \sim h\} = G_X g \subset \mathcal{F}(X)$, for $g \in \mathcal{F}(X)$, and the quotient set is denoted $\mathcal{F}(X)/G_X$.

**Theorem 3.** *Equivariant frame $\mathcal{F}(X)$ is a disjoint union of equal size orbits, $[g] \in \mathcal{F}(X)/G_X$.*

The proof is in A.3. The first immediate consequence of Theorem 3 is that the cardinality of $\mathcal{F}(X)$ is at-least that of the stabilizer (intuitively, the inner-symmetries) of $X$, namely $|\mathcal{F}(X)| \geq |G_X|$. Therefore, there could be cases, such as when $X$ describes a symmetric graph, where $|\mathcal{F}(X)|$ could be too large to average over. A remedy comes from the following observation: for every $h \in [g]$, we have that $h = rg$, $r \in G_X$, and $\phi(\rho_1(h)^{-1}X) = \phi(\rho_1(g)^{-1}\rho_1(r)^{-1}X) = \phi(\rho_1(g)^{-1}X)$, since also $r^{-1} \in G_X$. Therefore the summands in equations 2 are constant over orbits, and we get

$$\langle\phi\rangle_{\mathcal{F}}(X) = \frac{1}{m_{\mathcal{F}}} \sum_{[g] \in \mathcal{F}(X)/G_X} \phi(\rho_1(g)^{-1}X), \tag{6}$$

where $m_{\mathcal{F}} = |\mathcal{F}(X)/G_X| = |\mathcal{F}(X)|/|G_X|$. This representation of invariant FA requires only $m_{\mathcal{F}} = |\mathcal{F}(X)|/|G_X|$ evaluations, compared to $|\mathcal{F}(X)|$ in the original FA in equation 2.

**Approximation of invariant frame averaging.** Unfortunately, enumerating $\mathcal{F}(X)/G_X$ could be challenging in some cases. Nevertheless, equation 6 is still very useful: it turns out we can easily *draw a random element* from $\mathcal{F}(X)/G_X$ with uniform probability. This is an immediate application of the equal orbit size in Theorem 3:

**Corollary 1.** *Let $\mathcal{F}(X)$ be an equivariant frame, and $g \in \mathcal{F}(X)$ be a uniform random sample. Then $[g] \in \mathcal{F}(X)/G_X$ is also uniform.*

Therefore, an efficient approximation strategy is averaging over uniform samples, $g_i \in \mathcal{F}(X)$, $i \in [k]$,

$$\langle\!\langle\phi\rangle\!\rangle_{\mathcal{F}}(X) = \frac{1}{k} \sum_{i=1}^{k} \phi(\rho_1(g_i)^{-1}X). \tag{7}$$

This approximation is especially useful, compared to the full-blown FA, when $m_{\mathcal{F}} = |\mathcal{F}(X)|/|G_X|$ is small, i.e., when $|G_X|$ is large, or $X$ has many symmetries. Intuitively, the smaller $m_{\mathcal{F}}$ the better the approximation in equation 7. A partial explanation to this phenomenon is given in Appendix A.4, while an empirical validation is provided in Section 5.2.

## 2.2 EXPRESSIVE POWER

Another benefit in frame averaging as presented in equations 2 and 3 is that it preserves the expressive power of the base models, $\phi$, $\Phi$, as exaplined next. Consider some hypothesis function space $\mathcal{H} = \{\Phi\} \subset \mathcal{C}(V, W)$, where $\mathcal{C}(V, W)$ is the set of all continuous functions $V \to W$. As mentioned above, the case of scalar functions $\phi$ is a special case where $W = \mathbb{R}$, and $\rho_2(g) \equiv 1$. $\mathcal{H}$ can be seen as the collection of all functions represented by a certain class of neural networks, e.g., Multilayer Perceptron (MLP), or DeepSets (Zaheer et al., 2017), or Message Passing Graph Neural Networks (Gilmer et al., 2017). We denote by $\langle\mathcal{H}\rangle$ the collection of functions $\Phi \in \mathcal{H}$ after applying the frame averaging in equation 3, $\langle\mathcal{H}\rangle = \{\langle\Phi\rangle_{\mathcal{F}} \mid \Phi \in \mathcal{H}\}$.

We set some domain $K \subset V$ over which we would like to test the approximation power of $\langle \mathcal{H} \rangle$. To make sure that FA is well defined over $K$ we will assume it is *frame-finite*, i.e., for every $X \in K$, $\mathcal{F}(X)$ is a finite set. Next, we denote $K_{\mathcal{F}} = \left\{ \rho_1(g)^{-1} X \mid X \in K, g \in \mathcal{F}(X) \right\}$; intuitively, $K_{\mathcal{F}} \subset V$ contains all the points sampled by the FA operator. Lastly, to quantify approximation error over a set $A \subset V$ let us use the maximum norm $\|\Phi\|_{A,W} = \max_{X \in A} \|\Phi(X)\|_W$. We prove that an arbitrary equivariant function $\Psi \in \mathcal{C}(V, W)$ approximable by a function from $\mathcal{H}$ over $K_{\mathcal{F}}$ is approximable by an equivariant function from $\langle \mathcal{H} \rangle$ (Proof details are found on Appendix A.5).

**Theorem 4** (Expressive power of FA). *If $\mathcal{F}$ is a bounded $G$-equivariant frame, defined over a frame-finite domain $K$, then for an arbitrary equivariant function $\Psi \in \mathcal{C}(V, W)$ we have*

$$\inf_{\Phi \in \mathcal{H}} \|\Psi - \langle \Phi \rangle_{\mathcal{F}}\|_{K,W} \le c \inf_{\Phi \in \mathcal{H}} \|\Psi - \Phi\|_{K_{\mathcal{F}}, W},$$

*Where $c$ is the constant from Definition 1.*

This theorem can be used to prove universality results if the backbone model is universal, even for non-compact groups (e.g., the Euclidean motion group). Below we will use it to prove universality of frame averaged architectures for graphs, and point clouds with Euclidean motion symmetry.

## 3 MODEL INSTANCES

We instantiate the FA framework by specifying: i) The symmetry group $G$, representations $\rho_1, \rho_2$ and the underlying frame $\mathcal{F}$; ii) The backbone architecture for $\phi$ (invariant) or $\Phi$ (equivariant).

### 3.1 POINT CLOUDS, EUCLIDEAN MOTIONS.

**Symmetry.** We would like to incorporate Euclidean symmetry to existing permutation invariant/equivaraint point cloud networks. The symmetry of interest is $G = E(d) = O(d) \ltimes T(d)$, namely the group of Euclidean motions in $\mathbb{R}^d$ defined by rotations and reflections $O(d)$, and translations $T(d)$. We also discuss $G = SE(d) = SO(d) \ltimes T(d)$, where $SO(d)$ is the group of rotations in $\mathbb{R}^d$. We define $V = \mathbb{R}^{n \times d}$, and the group representation[2] is $\rho_1(g)X = XR^T + \mathbf{1}t^T$, where $R \in O(d)$ or $R \in SO(d)$, and $t \in \mathbb{R}^d$ denotes the translation. $W, \rho_2$ are defined similarly, unless translation invariance is desired in which case we use the representation $\rho_2(g)X = XR^T$.

**Frame.** We define the frame $\mathcal{F}(X)$ in this case based on Principle Component Analysis (PCA), as follows. Let $t = \frac{1}{n}X^T \mathbf{1} \in \mathbb{R}^d$ be the centroid of $X$, and $C = (X - \mathbf{1}t^T)^T (X - \mathbf{1}t^T) \in \mathbb{R}^{d \times d}$ the covariance matrix computed after removing the centroid from $X$. In the generic case the eigenvalues of $C$ satisfy $\lambda_1 < \lambda_2 < \cdots < \lambda_d$. Let $v_1, v_2, \ldots, v_d$ be the unit length corresponding eigenvectors. Then we define $\mathcal{F}(X) = \{([\alpha_1 v_1, \ldots, \alpha_d v_d], t) \mid \alpha_i \in \{-1, 1\}\} \subset E(d)$. The size of this frame (when it is defined) is $2^d$ which for typical dimensions $d = 2, 3$ amounts to frames of size $4, 8$, respectively. For $G = SE(d)$, we restrict $\mathcal{F}(X)$ to orthogonal, positive orientation matrices; generically there are $2^{d-1}$ such elements, which amounts to $2, 4$ elements for $d = 2, 3$, respectively.

**Proposition 1.** $\mathcal{F}(X)$ *based on the covariance and centroid are $E(d)$ equivariant and bounded.*

This choice of frame is defined for every $X \in V$ for which the covariance matrix $C$ has simple spectrum (i.e., non-repeating eigenvalues). It is known that within symmetric matrices, those with repeating eigenvalues are of co-dimension 2 (see e.g., Breiding et al. (2018)). Therefore, $\mathcal{F}(X)$ is defined for almost all $\mathcal{X}$ except rare singular points. Where it is defined, $\mathcal{F}$ is continuous as a direct consequence of perturbation theory of eigenvalues and eigenvectors of normal matrices (see e.g., Theorem 8.1.12 in Golub & Van Loan (1996)). However, when very close to repeating eigenvalues, small perturbation can lead to large frame change. In Appendix B we present an empirical study of frame stability and likelihood of encountering repeating eigenvalues in practice.

Since we would like to incorporate $E(d)$ symmetries to an already $S_n$ invariant/equivariant architectures, per Theorem 2, we will also need to show that the $\rho_1$ (and similarly $\rho_2$) commute with $\tau : S_n \to \mathrm{GL}(V)$ defined by $\tau(h)X = PX$, where $P = P_h$, $h \in S_n$, is the permutation representation. That is $P_h \in \mathbb{R}^{n \times n}$ is the permutation matrix representing $h \in S_n$, that is, $P_{ij} = 1$ if $i = h(j)$ and 0 otherwise. Indeed $\tau(h)\rho_1(g)X = P(XR^T + \mathbf{1}t^T) = \rho_1(g)\tau(h)X$. Furthermore, note that $\mathcal{F}(\tau(h)X) = \mathcal{F}(X)$, therefore $\mathcal{F}$ is also $S_n$ invariant.

---

[2]Technically, this representation is defined by matrices $\begin{pmatrix} R & t \\ \mathbf{0}^T & 1 \end{pmatrix}$ acting on $X' = [X, \mathbf{1}] \in \mathbb{R}^{n \times (d+1)}$.

**Backbone architectures.** We incorporate the FA framework with two existing popular point cloud network layers: i) PointNet (Qi et al., 2017a); and ii) DGCNN (Wang et al., 2018). We denote both architectures by $\Phi_{d,d'} : \mathbb{R}^{n \times d} \to \mathbb{R}^{n \times d'}$ for the $S_n$ equivariant version of these models. To simplify the discussion, we omit particular choices of layers and feature dimensions, Appendix C.1 provides the full details. We experimented with two design choices: First, consider frame averaged $\Phi_{3,3}$, i.e., $\langle \Phi_{3,3} \rangle_{\mathcal{F}}$, yielding a universal, $E(3)$ equivariant versions for PointNet and DGCNN, dubbed FA-PointNet and FA-DGCNN. A more complex design choice, taking inspiration from deep architectures with multiple modules, is to compose blocks of FA networks. For example, to build a *local $E(3)$* equivariant version of PointNet, denote also $\Upsilon_{d,d'} : \mathbb{R}^d \to \mathbb{R}^{d'}$ an MLP. We decompose the input point cloud to $k$-nn patches $\boldsymbol{X}_i \in \mathbb{R}^{k \times 3}$, $i \in [n]$, where each patch is sorted by distance. Next, feed each patch into an equivariant FA $\Upsilon_{3k,3d}$, each with its own frame $\mathcal{F}_i$, resulting in $E(3)$ equivariant features in $\mathbb{R}^{3d}$ for every point, $\boldsymbol{Y} \in \mathbb{R}^{n \times 3d}$; then applying equivariant FA $\Phi_{3d,3}$ providing output in $\mathbb{R}^{n \times 3}$. That is, $\Psi(\boldsymbol{X}) = \langle \Phi_{3d,3} \rangle_{\mathcal{F}} \big( \big[ \langle \Upsilon_{3k,3d} \rangle_{\mathcal{F}_1}(\boldsymbol{X}_1), \dots, \langle \Upsilon_{3k,3d} \rangle_{\mathcal{F}_n}(\boldsymbol{X}_n) \big] \big)$, where brackets denote concat in the first dimension. We name this construction FA-Local-PointNet.

**Universality.** We use Theorem 4 to prove that using a universal set invariant/equivariant backbone $\phi, \Phi$, such as DeepSets or PointNet (see e.g., Zaheer et al. (2017); Qi et al. (2017a); Segol & Lipman (2019)) leads to a universal model. Let $\mathcal{H}$ be any such universal set-equivariant function collection. That is, for arbitrary continuous set function $\Psi$ we have (in the notation of Section 2.2) $\inf_{\Phi \in \mathcal{H}} \|\Psi - \Phi\|_{\Omega,W} = 0$ for arbitrary compact sets $\Omega \subset V$. If $K \subset V$ is some bounded domain, then the choice of frame $\mathcal{F}$ described above implies that $K_{\mathcal{F}}$ is also bounded and therefore contained in some compact set $\Omega \subset V$. Consequently, Proposition 1 and Theorem 4 imply Corollary 2. A similar results holds for $SE(d)$, which provides a similar expressiveness guarantee as the one from (Dym & Maron, 2020) analyzing Tensor Field Networks (Thomas et al., 2018; Fuchs et al., 2020).

**Corollary 2** (FA-DeepSets/PointNet are universal). *Frame Averaging DeepSets/PointNet using the frame $\mathcal{F}$ defined above results in a universal $E(d) \times S_n$ invariant/equivariant model over bounded frame-finite sets $K \subset V$.*

## 3.2 GRAPHS, PERMUTATIONS

**Symmetry and frame.** Let $G = S_n$, and $V = \mathbb{R}^{n \times d} \times \mathbb{R}^{n \times n}$, where $\mathbf{X} = (\boldsymbol{Y}, \boldsymbol{A}) \in V$ represents a set of node features $\boldsymbol{Y} \in \mathbb{R}^{n \times d}$, and an adjacency matrix (or some edge attributes) $\boldsymbol{A} \in \mathbb{R}^{n \times n}$; we assume undirected graphs, meaning $\boldsymbol{A} = \boldsymbol{A}^T$. Let $\mathbf{X} \in V$, the representation $\rho_1$ is defined by $\rho_1(g)\mathbf{X} = (\boldsymbol{P}\boldsymbol{Y}, \boldsymbol{P}\boldsymbol{A}\boldsymbol{P}^T)$, where $\boldsymbol{P} = \boldsymbol{P}_g$ is the permutation matrix representing $g \in S_n$. We define $\mathcal{F}(\mathbf{X})$ to contain all $g \in S_n$ that sort the rows of the matrix $\boldsymbol{S}(\mathbf{X})$ in column lexicographic manner, i.e., $\boldsymbol{P}\boldsymbol{S}(\mathbf{X})$ is lexicographically sorted; the matrix $\boldsymbol{S}$ is defined as follows. Let $\boldsymbol{L} = \mathrm{diag}(\boldsymbol{A}\boldsymbol{1}) - \boldsymbol{A}$ be the graph's Laplacian. For every eigenspace (traversed in increasing eigenvalue order), spanned by the orthogonal basis $\boldsymbol{u}_1, \dots, \boldsymbol{u}_k$, we add the (equivariant) column $\mathrm{diag}(\sum_{i=1}^k \boldsymbol{u}_i \boldsymbol{u}_i^T)$ to $\boldsymbol{S}$ (Fürer, 2010). Hence, the number of columns of $\boldsymbol{S}$ equals to the number of unique eigenvalues of $\boldsymbol{L}$.

**Proposition 2.** $\mathcal{F}(\mathbf{X})$ *defined by sorting of $\boldsymbol{S}(\mathbf{X})$ is $S_n$-equivariant and bounded.*

**Backbone architectures.** In this case we perform only invariant tasks, for which we chose two universal backbone architectures for $\phi$: (i) MLP applied to $\mathrm{vec}(\boldsymbol{Y}, \boldsymbol{A})$ and (ii) GNN+ID (Murphy et al., 2019; Loukas, 2020), denoting a GNN backbone equipped with node identifiers as node features. We perform FA with $\phi$ according to the frame constructed in Proposition 2. Note that Theorem 3 implies, $|\mathcal{F}(X)| \geq |\mathrm{Aut}(G)|$, since the stabilizer $G_{\mathbf{X}}$ is the automorphism group of the graph, i.e., $g \in G_{\mathbf{X}} = \mathrm{Aut}(\mathbf{X})$ iff $\rho_1(g)\mathbf{X} = \mathbf{X}$. This means that for symmetric graphs equation 2 can prove too costly. In this case, we use the approximate invariant FA, equation 7.

**Universality.** Let us use Theorem 4 to prove FA-MLP and FA-GNN+ID are universal. Again, it is enough to consider the equivariant case. Let $\mathcal{H} \subset \mathcal{C}(V, W)$ denote the collection of functions that can be represented by MLPs or GNN+ID. Universality results in (Pinkus, 1999; Loukas, 2020; Puny et al., 2020) imply that for any continuous graph (i.e., $S_n$ equivariant) function $\Psi$, $\inf_{\Phi \in \mathcal{H}} \|\Psi - \Phi\|_{\Omega,W} = 0$ for any compact $\Omega \subset V$. Let $K \subset V$ be some bounded domain, then since $G$ is finite then $K_{\mathcal{F}}$ is also bounded and is contained is some compact set. Proposition 2 and Theorem 4 now imply:

**Corollary 3** (FA-MLP and FA-GNN+ID is graph universal). *Frame Averaging MLP/GNN+ID using the frame $\mathcal{F}$ above results in a universal $S_n$ equivariant graph model over bounded domains $K \subset V$.*

### 3.3 Graphs, Euclidean motions.

**Symmetry and frame.** We consider the group $G = E(d)$ acting on graphs, i.e., $V = \mathbb{R}^{n \times d} \times \mathbb{R}^{n \times n}$, where $\mathbf{X} = (\boldsymbol{Y}, \boldsymbol{A}) \in V$ represents a set of node and edge attributes, as described above. The group representation is $\rho_1(g)\mathbf{X} = \rho_1(g)(\boldsymbol{Y}, \boldsymbol{A}) = (\boldsymbol{Y}\boldsymbol{R}^T + \mathbf{1}\boldsymbol{t}^T, \boldsymbol{A})$. We define the frame $\mathcal{F}(\mathbf{X}) = \mathcal{F}(\boldsymbol{Y})$ using the node features as in the point cloud case, Section 3.1. Therefore Proposition 1 implies $\mathcal{F}$ is equivariant and bounded. Next, also in this case we would like to incorporate $E(d)$ symmetries to an already $S_n$ invariant/equivariant graph neural network architectures; again per Theorem 2, we will also need to show that the $\rho_1$ (and similarly $\rho_2$) commutes with $\tau : S_n \to \mathrm{GL}(V)$ defined by $\tau(h)\mathbf{X} = (\boldsymbol{P}\boldsymbol{Y}, \boldsymbol{P}\boldsymbol{A}\boldsymbol{P}^T)$, where $\boldsymbol{P} = \boldsymbol{P}_h$ is the permutation matrix of $h \in S_n$. Indeed $\tau(h)\rho_1(g)\mathbf{X} = (\boldsymbol{P}(\boldsymbol{Y}\boldsymbol{R}^T + \mathbf{1}\boldsymbol{t}^T), \boldsymbol{P}\boldsymbol{A}\boldsymbol{P}^T) = \rho_1(g)\tau(h)\mathbf{X}$. Furthermore, note that as in the point cloud case $\mathcal{F}(\tau(h)\mathbf{X}) = \mathcal{F}(\mathbf{X})$, therefore $\mathcal{F}$ is also $S_n$ invariant.

**Backbone architecture.** The backbone architecture we chose for this instantiation is the Message Passing GNN in Gilmer et al. (2017), an $S_n$ equivariant model denoted $\Phi_{d,d'} : \mathbb{R}^{n \times d} \times \mathbb{R}^{n \times n} \to \mathbb{R}^{n \times d'} \times \mathbb{R}^{n \times n}$. In this case we constructed a model, as suggested above, by composing $l$ equivariant layers $\Psi(\mathbf{X}) = \langle \Phi_{3d',3}^{(l)} \rangle_{\mathcal{F}} \circ ... \langle \Phi_{3d',3d'}^{(i)} \rangle_{\mathcal{F}} ... \circ \langle \Phi_{6,3d'}^{(1)} \rangle_{\mathcal{F}}(\mathbf{X})$. The input feature size is 6 since we use velocities in addition to initial position as input ($n$-body problem). We name this model FA-GNN.

## 4 Previous Works

**Rotation invariant and equivariant point networks.** State of the art $S_n$ invariant networks, e.g., (Qi et al., 2017a;b; Atzmon et al., 2018; Li et al., 2018; Xu et al., 2018b; Wang et al., 2018) are not invariant/equivariant to rotations/reflections by construction (Chen et al., 2019). Invariance to global or local rotations can be achieved by modifying the 3D convolution operator or modifying the input representation. Relative angles and distances across points (Deng et al., 2018; Zhang et al., 2019) or angles and distances w.r.t. normals (Gojcic et al., 2019) can be used for rotation invariance. Other works use some local or global frames to achieve invariance to rotations and translations. Xiao et al. (2020); Yu et al. (2020); Deng et al. (2018) also use PCA to define rotation invariance, and can be seen as instances of the FA framework. We augment this line of work by introducing a more general framework that includes equivariance to rotation/reflection and translation, more general architectures and symmetries as well as theoretical analysis of the expressive power of such models.
Equivariance is a desirable property for 3D recognition, registration (Ao et al., 2021), and other domains such as complex physical systems (Kondor, 2018). A popular line of work utilizes the theory of spherical harmonics to achieve equivariance (Worrall et al., 2017; Esteves et al., 2018; Liu et al., 2018; Weiler et al., 2018; Cohen et al., 2018). Notably, Tensor Field Networks (TFN), $SE(3)$ transformers, and Group-Equivariant Attention (Thomas et al., 2018; Fuchs et al., 2020; Romero & Cordonnier, 2021) achieve equivariance to both translation and rotation, i.e., $SE(3)$, and are maximally expressive (i.e., universal) as shown in Dym & Maron (2020). These methods, however, are specifically adapted to $SE(3)$ and require high order $SO(3)$ representations as features. Recently Deng et al. (2021) propose a rotation equivariant network by introducing tensor features, linear layers that act equivariantly on them and equivariant non-linearities etc. However, their architecture is not proved to be universal. Discrete convolutions (Cohen et al., 2019; Li et al., 2019; Worrall & Brostow, 2018) have also been used for achieving equivariance. In particular, Chen et al. (2021) propose point networks that are $SE(3)$ equivariant and use separable discrete convolutions. Lastly, (Finzi et al., 2020) construct equivariant layers using local group convolution, and extends beyond rotations to any Lie group.

**Graph neural networks.** Message passing (MP) GNNs (Gilmer et al., 2017) are designed to be $S_n$ equivariant. Kondor et al. (2018) introduces a broader set of equivariant operators in MP-GNNs, while Maron et al. (2018) provides a full characterization of linear permutation invariant/equivariant GNN layers. In a parallel approach, trying to avoid harming expressively due to restricted architectures (Xu et al., 2018a; Morris et al., 2019), other works suggested symmetrization of non invariant/equivariant backbones. Ranging from eliminating all symmetries by a canonical ordering (Niepert et al., 2016) to averaging over the entire symmetry group (Murphy et al., 2019; 2018), which amount to the trivial frame $\mathcal{F} \equiv \rho(G)$, with the symmetry group $G = S_n$. As we have also shown, this approach comes at a cost of high variance approximations hindering the learning process. Our FA framework reduces the variance both by choosing a canonical ordering and addressing the fact that it may not be unique.

Recently, a body of work studies GNNs with invariance/equivariance to $E(3)$ (or a similar group) to deal with symmetries in molecular data or dynamical systems . Many $SE(3)$ equivariant con-

structions (Anderson et al., 2019; Fuchs et al., 2020; Batzner et al., 2021; Klicpera et al., 2021) extend TFN (Thomas et al., 2018) and inherit its expensive higher order feature representations in the intermediate layers. Finally, a recent work by Satorras et al. (2021) provides an efficient message passing construction which is $E(d)$ equivariance but is not shown to be universal, thus far.

## 5 Experiments

We evaluate our FA framework on a few invariant/equivaraint point cloud and graph learning tasks: point cloud normal estimation ($O(3)$ equivariant and translation invariant); graph separation tasks ($S_n$ invariant); and particles position estimation, i.e., the $n$-body problem ($E(3)$ equivariant).

### 5.1 Point Clouds: Normal Estimation

Normal estimation is a core *geometry processing* task, where the goal is to estimate normal data from an input point cloud, $\boldsymbol{X} \in \mathbb{R}^{n \times 3}$. This task is $O(3)$ equivariant and translation invariant. To test the effect of rotated data on the different models we experimented with three different settings: i) $I/I$ - training and testing on the original data; ii) $I/SO(3)$ - training on the original data and testing on randomly rotated data; and iii) $SO(3)/SO(3)$ - training and testing with randomly rotated data. We used the ABC dataset (Koch et al., 2019) that contains 3 collections (10k, 50k, and 100k models each) of *Computer-Aided Design* (CAD) models. We follow the protocol of the benchmark suggested in Koch et al. (2019), and quantitatively measure normal estimation quality via $1 - (\boldsymbol{n}^T \hat{\boldsymbol{n}})^2$, with $\boldsymbol{n}$ being the ground truth normal and $\hat{\boldsymbol{n}}$ the normal prediction. We used the same random train/test splits from Koch et al. (2019). For baselines, we chose the PointNet (Qi et al., 2017a) and DGCNN (Wang et al., 2018) architectures, which are popular permutation equivariant 3D point cloud architectures. In addition, we also compare to VN-PointNet and VN-DGCNN (Deng et al., 2021), a recent state of the art approach for $SO(3)$ equivariant network design. We also tested our FA models, FA-PointNet and FA-DGCNN as described in Section 3.1. In addition, we tested our local $O(3)$ equivariant model, FA-Local-PointNet. See Appendix C.1 for the further implementation details. The results in Table 1 showcase the advantages of our FA framework: Incorporating FA to existing architectures is beneficial in scenarios (ii-iii), outperforming augmentation by a large margin. In contrast to VN models, FA models maintain (and in some cases even improve) the baseline estimation quality (i), attributed to the expressive power of the FA models.

| Method | 10k | | | 50k | | | 100k | | |
|---|---|---|---|---|---|---|---|---|---|
| | $I/I$ | $I/SO(3)$ | $SO(3)/SO(3)$ | $I/I$ | $I/SO(3)$ | $SO(3)/SO(3)$ | $I/I$ | $I/SO(3)$ | $SO(3)/SO(3)$ |
| PointNet | .207±.004 | .449±.006 | .258±.002 | .188±.002 | .430±.007 | .232±.001 | .188±.006 | .419±.006 | .231±.001 |
| VN-PointNet | .215±.003 | .216±.004 | .223±.004 | .185±.002 | .186±.006 | .187±.006 | .189±.004 | .188±.007 | .185±.003 |
| FA-PointNet | .158±.001 | .163±.001 | .161±.002 | .148±.001 | .148±.003 | .150±.002 | .148±.002 | .147±.001 | .149±.001 |
| FA-Local-PointNet | .097±.001 | .098±.001 | .098±.001 | .091±.001 | .090±.001 | .091±.001 | .091±.001 | .090±.002 | .091±.002 |
| DGCNN | .070±.003 | .193±.015 | .121±.001 | **.061±.004** | .174±.007 | .122±.001 | **.058±.002** | .173±.003 | .112±.001 |
| VN-DGCNN | .133±.003 | .130±.001 | .144±.007 | .127±.005 | .125±.001 | .127±.006 | .127±.005 | .125±.001 | .127±.005 |
| FA-DGCNN | **.067±.001** | **.069±.002** | **.071±.003** | .065±.001 | **.067±.004** | **.068±.004** | .073±.008 | **.067±.001** | **.071±.009** |

Table 1: Normal estimation, ABC dataset (Koch et al., 2019) benchmark.

### 5.2 Graphs: Expressive Power

Producing GNNs that are both expressive and computationally tractable is a long standing goal of the graph learning community. In this experiment we test graph separation ($S_n$ invariant task): the ability of models to separate and classify graphs, a basic trait for graph learning. We use two datasets: GRAPH8c (Balcilar et al., 2021) that consists of all non-isomorphic, connected 8 node graphs; and EXP (Abboud et al., 2021) that consists of 3-WL distinguishable graphs that are not 2-WL distinguishable. There are two tasks: (i) count pairs of graphs not separated by a randomly initialized model in GRAPH8c and EXP; and (ii) learn to classify EXP to two classes. We follow Balcilar et al. (2021) experimental setup. As baselines we use GCN Kipf & Welling (2016), GAT Veličković et al. (2018), GIN Xu et al. (2018a), CHEBNET Tang et al.

| Model | GRAPH8c | EXP | EXP-classify |
|---|---|---|---|
| GCN | 4755 | 600 | 50% |
| GAT | 1828 | 600 | 50% |
| GIN | 386 | 600 | 50% |
| CHEBNET | 44 | 71 | 82% |
| PPGN | 0 | 0 | **100%** |
| GNNML3 | 0 | 0 | **100%** |
| GA-MLP | 0 | 0 | 50% |
| FA-MLP | 0 | 0 | **100%** |
| GA-GIN+ID | 0 | 0 | 50% |
| FA-GIN+ID | 0 | 0 | **100%** |

Table 2: Graph separation (Balcilar et al., 2021; Abboud et al., 2021).

(2019), PPGN Maron et al. (2019), and GNNML3 Balcilar et al. (2021), all of which are equivariant by construction. We compare to our FA-MLP, and FA-GIN+ID, as described in Section 3.2 that provide tractable option for universal GNNs. We also compare to the trivial (i.e., entire group) frame averaging $\mathcal{F} \equiv G$, denoted GA-MLP and GA-GIN+ID as advocated in (Murphy et al., 2019). Appendix C.2 provides full implementation details.

The two first columns in Table 2 show the results of the separation task (i). As expected both FA and GA provide perfect separation, however, are not perfectly invariant unless one computes the full averages on $\mathcal{F}$ and $G$, respectively. Since for some graphs (with many symmetries) full averages are not feasible, we compare the *invariance error* (see appendix for the exact definition) of GA with that of the approximate FA, equation 7, for 50 randomly permuted inputs. Figure 2-left shows the mean, std, and $90\%$ percentile of *invariance error* for increasing sample size $k$. Note that approximate FA is more invariant

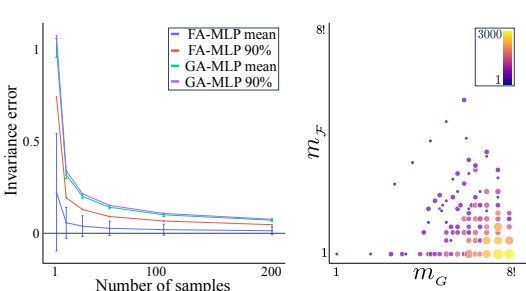

Figure 2: Invariance error (left); $m_{\mathcal{F}}$-$m_G$ (right).

even with as little as $k = 1$ samples, which explains why in the classification task (ii), in Table 2, right column, FA is able to learn while GA fails, when both are trained with sample size $k = 1$. Figure 2-right compares $m_{\mathcal{F}}$ and $m_G$ (the maximal number of unique elements in the FA, equation 6) for GRAPH8c dataset. The color and size of the points in the plot represents how many graphs in the dataset have the corresponding $(m_G, m_{\mathcal{F}})$ values. This demonstrates the benefit (in one example) of FA over GA in view of Theorem 5 and approximation in equation 7. Of course, other, more powerful frames can be chosen, e.g., using higher order WL (Morris et al., 2019) or substructure counting (Bouritsas et al., 2020) that will further improve the approximation of equation 7.

## 5.3 GRAPHS: $n$-BODY PROBLEM

In this task we learn to solve the $n$-body problem ($E(3)$ equivariant). The dataset, created in (Satorras et al., 2021; Fuchs et al., 2020), consists of a collection of $n = 5$ particles systems, where each particle is equipped with its initial position in $\mathbb{R}^3$ and velocity in $\mathbb{R}^3$, and each pair of particles (edge) with a value indicating their charge difference. The task is to predict the particles' locations after a fixed time. We follow the protocol of (Satorras et al., 2021). As baselines we use EGNN (Satorras et al., 2021), GNN (Gilmer et al., 2017), TFN (Thomas et al., 2018), SE(3)-Transformer (Fuchs et al., 2020) and Radial Field (Köhler et al., 2019). We test our FA-GNN, as described in Section 3.3. Table 3 logs the results, where the metric reported

| Method | MSE | Forward time (s) |
|---|---|---|
| Linear | 0.0819 | .0001 |
| SE(3) Transformer | 0.0244 | .1346 |
| TFN | 0.0155 | .0343 |
| GNN | 0.0107 | .0032 |
| Radial Field | 0.0104 | .0039 |
| EGNN | 0.0071 | .0062 |
| FA-GNN | **0.0057** | .0041 |

Table 3: $n$-body experiment (Satorras et al., 2021).

is the Mean Squared Error between predicted locations and ground truth. As can be seen, FA-GNN improves over the SOTA by more than 20% . Note that the number of parameters used in FA-GNN and EGNN is roughly the same. More details are provided in Appendix C.3

## 6 CONCLUSIONS

We present Frame Averaging, a generic and principled methodology for adapting existing (backbone) neural architectures to be invariant/equivariant to desired symmetries that appear in the data. We prove the method preserves the expressive power of the backbone model, and is efficient to compute in several cases of interest. We use FA to build universal GNNs, universal point cloud networks that are invariant/equivariant to Euclidean motions $E(3)$, and GNNs invariant/equivariant to Euclidean motions $E(3)$. We empirically validate the effectiveness of these models on several invariant/equivariant learning tasks. We believe the instantiations presented in the paper are only the first step in exploring the full potential of the FA framework, and there are many other symmetries and scenarios that can benefit from FA. For example, extending the invariance (or equivariance) of a model from a subgroup $H$ of $G$, to $G$. Further Interesting open questions, not answered by this paper are: What would be a way to systematically find efficient/small frames? How the frame choice effects the learning process? How to explore useful FA architectures and modules?

ACKNOWLEDGMENTS

OP, MA and HB were supported by the European Research Council (ERC Consolidator Grant, "LiftMatch" 771136) and also by a research grant from the Carolito Stiftung (WAIC).

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

# A  PROOFS

## A.1  PROOF OF THEOREM 1

*Proof.* First note that frame equivariance is defined to be $\mathcal{F}(\rho_1(g)X) = g\mathcal{F}(X)$ which in particular means $|\mathcal{F}(\rho_1(g)X)| = |\mathcal{F}(X)|$.

For invariance, let $g' \in G$

$$\langle\phi\rangle_{\mathcal{F}}(\rho_1(g')X) = \frac{1}{|\mathcal{F}(X)|}\sum_{g\in g'\mathcal{F}(X)}\phi(\rho_1(g)^{-1}\rho_1(g')X) = \frac{1}{|\mathcal{F}(X)|}\sum_{g\in\mathcal{F}(X)}\phi(\rho_1(g'g)^{-1}\rho_1(g')X)$$

$$= \frac{1}{|\mathcal{F}(X)|}\sum_{g\in\mathcal{F}(X)}\phi(\rho_1(g)^{-1}X) = \langle\phi\rangle_{\mathcal{F}}(X)$$

and for equivariance

$$\langle\Phi\rangle_{\mathcal{F}}(\rho_1(g')X) = \frac{1}{|\mathcal{F}(X)|}\sum_{g\in g'\mathcal{F}(X)}\rho_2(g)\Phi(\rho_1(g)^{-1}\rho_1(g')X)$$

$$= \frac{1}{|\mathcal{F}(X)|}\sum_{g\in\mathcal{F}(X)}\rho_2(g'g)\Phi(\rho_1(g'g)^{-1}\rho_1(g')X)$$

$$= \rho_2(g')\frac{1}{|\mathcal{F}(X)|}\sum_{g\in\mathcal{F}(X)}\rho_2(g)\Phi(\rho_1(g)^{-1}X)$$

$$= \rho_2(g')\langle\Phi\rangle_{\mathcal{F}}(X)$$

$\square$

## A.2  PROOF OF THEOREM 2

*Proof.* The proof above A.1, of Theorem 1, is a special case of the following where the second symmetry is chosen to be trivial. In principle the proofs are quite similar.

First note that frame equivariance and invariance, together with $\rho_1, \tau_1$ commuting mean

$$\mathcal{F}(\gamma_1(h,g)X) = \mathcal{F}(\tau_1(h)\rho_1(g)X) = \mathcal{F}(\rho_1(g)\tau_1(h)X) = g\mathcal{F}(\tau_1(h)X) = g\mathcal{F}(X),$$

which in particular implies that $|\mathcal{F}(\gamma_1(h,g)X)| = |\mathcal{F}(X)|$. Let $(h',g') \in H \times G$ be arbitrary. Then,

$$\langle\phi\rangle_{\mathcal{F}}(\gamma_1(h',g')X) = \frac{1}{|\mathcal{F}(X)|}\sum_{g\in g'\mathcal{F}(X)}\phi(\rho_1(g)^{-1}\tau_1(h')\rho_1(g')X)$$

$$= \frac{1}{|\mathcal{F}(X)|}\sum_{g\in\mathcal{F}(X)}\phi(\rho_1(g'g)^{-1}\tau_1(h')\rho_1(g')X)$$

$$= \frac{1}{|\mathcal{F}(X)|}\sum_{g\in\mathcal{F}(X)}\phi(\rho_1(g)^{-1}\tau_1(h')X)$$

$$= \frac{1}{|\mathcal{F}(X)|}\sum_{g\in\mathcal{F}(X)}\phi(\rho_1(g)^{-1}X)$$

$$= \langle\phi\rangle_{\mathcal{F}}(X)$$

meaning that $\langle \phi \rangle_{\mathcal{F}}$ is $H \times G$ invariant. Next,

$$
\begin{aligned}
\langle \Phi \rangle_{\mathcal{F}} (\gamma_1(h', g')X) &= \frac{1}{|\mathcal{F}(X)|} \sum_{g \in g'\mathcal{F}(X)} \rho_2(g)\Phi(\rho_1(g)^{-1}\tau_1(h')\rho_1(g')X) \\
&= \frac{1}{|\mathcal{F}(X)|} \sum_{g \in \mathcal{F}(X)} \rho_2(g'g)\Phi(\rho_1(g'g)^{-1}\tau_1(h')\rho_1(g')X) \\
&= \frac{1}{|\mathcal{F}(X)|} \sum_{g \in \mathcal{F}(X)} \rho_2(g'g)\Phi(\rho_1(g)^{-1}\tau_1(h')X) \\
&= \tau_2(h')\rho_2(g')\frac{1}{|\mathcal{F}(X)|} \sum_{g \in \mathcal{F}(X)} \rho_2(g)\Phi(\rho_1(g)^{-1}X) \\
&= \gamma_2(h', g') \langle \Phi \rangle_{\mathcal{F}} (X)
\end{aligned}
$$

showing that $\langle \Phi \rangle_{\mathcal{F}}$ is $H \times G$ equivariant. $\qquad\square$

### A.3 PROOF OF THEOREM 3 AND COROLLARY 1

*Proof.* (Theorem 3) First we show $G_X$ acts on $\mathcal{F}(X)$. For an arbitrary $h \in G_X$ and equivariant frame $\mathcal{F}$, we have $\mathcal{F}(X) = \mathcal{F}(\rho_1(h)X) = h\mathcal{F}(X)$, where in the first equality we used the fact that $h \in G_X$ and in the second the equivariance of $\mathcal{F}$. This means that if $g \in \mathcal{F}(X)$ and $h \in G_X$ then also $hg \in \mathcal{F}(X)$, or in other words $\mathcal{F}(X)$ is closed to actions from $G_X$. Furthermore, since a group acts on itself via bijections we have that the cardinality of all orbits is the same, $|[g]| = |G_X g| = |G_X|$, for all $[g] \in \mathcal{F}(X)/G_X$. Lastly, the equivalence relation $g \sim h \iff hg^{-1} \in G_X$ shows that $\mathcal{F}(X)$ is a union of disjoint orbits of equal cardinality. $\qquad\square$

*Proof.* (Corollary 1) Theorem 3 asserts that all orbits $[g] \in \mathcal{F}(X)/G_X$ have the same number of elements. Therefore, a random choice $g \in \mathcal{F}(X)$ will have equal probability to land in each orbit. $\quad\square$

### A.4 THE ROLE OF $m_{\mathcal{F}}$ IN APPROXIMATION QUALITY OF EQUATION 7

To better understand the role of $m_{\mathcal{F}}$ in approximating equation 6 we first note that equation 7 can be written as $\langle\!\langle \phi \rangle\!\rangle_{\mathcal{F}} (X) = \sum_{[g] \in \mathcal{F}(X)/G_X} \hat{\mu}_{[g]} \phi(\rho_1(g_i)^{-1}X)$, where $\hat{\mu}$ is the empiricial distribution over $\mathcal{F}(X)/G_X$, assigning to each element $[g] \in \mathcal{F}(X)/G_X$ the fraction of samples landed in $[g]$, i.e., $\hat{\mu}_{[g]} = |\{i \in [k]|g_i \in [g]\}|k^{-1}$. We next present a lower bound on the probability of any particular $\hat{\mu}$ that provides a good approximation $\langle\!\langle \phi \rangle\!\rangle_{\mathcal{F}} (X) \approx \langle \phi \rangle_{\mathcal{F}} (X)$.

**Theorem 5.** *Let $\mathcal{F}$ be an equivariant frame. The probability of an arbitrary $\hat{\mu} \in \left\{ \hat{\mu} \mid \sup_{\phi \in \mathcal{Q}} |\langle \phi \rangle_{\mathcal{F}} (X) - \langle\!\langle \phi \rangle\!\rangle_{\mathcal{F}} (X)| \leq \epsilon \right\}$ is bounded from below as follows,*

$$
P(\hat{\mu}) \geq (1+k)^{-m_{\mathcal{F}}} \exp(-2m_{\mathcal{F}} k\epsilon^2),
$$

*where $\mathcal{Q} = \{\phi \in \mathcal{C}(V, \mathbb{R}) \mid |\phi(X)| \leq 1, \forall X \in V\}$ the set of bounded, continuous functions $V \to \mathbb{R}$.*

Before providing the proof let us note that this theorem provides a lower bound for each particular "good" empirical distribution $\hat{\mu}$. The main takeoff is that for fixed $k$ and $\epsilon$, the smaller $m_{\mathcal{F}}$ the better the lower bound. The counter-intuitive behaviour of this bound w.r.t. $k$ and $\epsilon$ stems from the fact that the size of the set of "good" $\hat{\mu}$, namely $\{\phi \in \mathcal{C}(V, \mathbb{R}) \mid |\phi(X)| \leq 1, \forall X \in V\}$ is increasing with $k$ and $\epsilon$.

*Proof.* For brevity we denote $m = m_{\mathcal{F}}$. Our setting can be formulated as follows. We have the uniform probability distribution, denoted $\mu$, over the discrete space $[m]$ (representing the quotient set $\mathcal{F}(X)/G_X$); $\mu_j = \frac{1}{m}$, $j \in [m]$, and we have numbers $a_j = \phi(\rho_1(h_j)^{-1}X)$, where $[h_j] \in \mathcal{F}(X)/G_X$, $j \in [m]$ represent exactly one sample per orbit. In this notation $\langle \phi \rangle_{\mathcal{F}} (X) = \sum_{j=1}^{m} \frac{a_j}{m}$, while the approximation in equation 7 takes the form $\langle\!\langle \phi \rangle\!\rangle_{\mathcal{F}} (X) = \sum_{j=1}^{m} \hat{\mu}_j a_j$, where $\hat{\mu}_j = k_j/k$, where $k_j$ is the number of samples $g_i \in [h_j]$, $i \in [k]$, i.e., samples that landed in the $j$-th orbit. Therefore,

$$
\sup_{\phi \in \mathcal{Q}} |\langle \phi \rangle_{\mathcal{F}} (X) - \langle\!\langle \phi \rangle\!\rangle_{\mathcal{F}} (X)| = \sup_{|a_j| \leq 1} \left| \sum_{j=1}^{m} a_j \left( \frac{1}{m} - \hat{\mu}_j \right) \right| = \sum_{j=1}^{m} \left| \frac{1}{m} - \hat{\mu}_j \right| = 2 \|\mu - \hat{\mu}\|_{\mathrm{TV}},
$$

where the latter is the total variation norm for discrete measures (Levin & Peres, 2017). We denote $H(\hat\mu|\mu) = \sum_{j=1}^m \log(\hat\mu_j) \log\left(\frac{\hat\mu_j}{\mu_j}\right)$ the KL-divergence. The Pinsker inequality and its inverse for discrete positive measures are (see e.g., (Polyanskiy & Wu, 2014)):

$$\frac{1}{2} \|\hat\mu - \mu\|_{\mathrm{TV}}^2 \le H(\hat\mu|\mu) \le \frac{2}{\alpha} \|\hat\mu - \mu\|_{\mathrm{TV}}^2, \tag{8}$$

where $\alpha = \min_{j\in[m]} \mu_j = m^{-1}$. Therefore,

$$\Gamma_\epsilon = \left\{ \hat\mu \;\Big|\; \sup_{\phi\in\mathcal{Q}} |\langle\phi\rangle_\mathcal{F}(X) - \langle\!\langle\phi\rangle\!\rangle_\mathcal{F}(X)| \le \epsilon \right\} = \left\{ \hat\mu \;\Big|\; 2\|\hat\mu - \mu\|_{\mathrm{TV}} \le \epsilon \right\} \subset \left\{ \hat\mu \;\Big|\; H(\bar\mu|\mu) \le \frac{\epsilon^2 m}{2} \right\}$$

Now, application of Large Deviation Theory (Lemma 2.1.9 in Dembo & Zeitouni (2010)) provides that for $\hat\mu$ so that $H(\hat\mu|\mu) \le \frac{\epsilon^2 m}{2}$:

$$P(\hat\mu) \ge \frac{1}{(1+k)^m} e^{-\frac{km\epsilon^2}{2}}.$$

$\square$

## A.5 Proof of Theorem 4

*Proof.* Let $\Psi \in \mathcal{C}(V, W)$ be an arbitrary $G$ equivariant function, $\mathcal{F}$ a bounded $G$ equivariant frame over a frame-finite domain $K$. Let $c > 0$ be the constant from Definition 1. For arbitrary $X \in K$,

$$
\begin{aligned}
\|\Psi(X) - \langle\Phi\rangle_\mathcal{F}(X)\|_W &= \|\langle\Psi\rangle_\mathcal{F}(X) - \langle\Phi\rangle_\mathcal{F}(X)\|_W \\
&\le \frac{1}{|\mathcal{F}(X)|} \sum_{g\in\mathcal{F}(X)} \left\| \rho_2(g)\Psi(\rho_1(g)^{-1}X) - \rho_2(g)\Phi(\rho_1(g)^{-1}X) \right\|_W \\
&\le \max_{g\in\mathcal{F}(X)} \|\rho_2(g)\|_{\mathrm{op}} \|\Psi - \Phi\|_{K_\mathcal{F}, W} \\
&\le c \|\Psi - \Phi\|_{K_\mathcal{F}, W}
\end{aligned}
$$

where in the first equality we used the fact that $\langle\Psi\rangle_\mathcal{F} = \Psi$ since $\Psi$ is already equivariant. $\square$

## A.6 Proof of Proposition 1

*Proof.* Let us prove $\mathcal{F}$ is equivariant (equation 3). Consider a transformation $g = (\boldsymbol{R}, \boldsymbol{t}) \in G$, and let $(\boldsymbol{O}, \boldsymbol{s}) \in \mathcal{F}(\boldsymbol{X})$, then

$$\boldsymbol{C} = \boldsymbol{O}\Lambda\boldsymbol{O}^T, \quad \boldsymbol{s} = \frac{1}{n}\boldsymbol{X}^T\mathbf{1},$$

where $\boldsymbol{O}\Lambda\boldsymbol{O}^T$ is the eigen decomposition of $C$. Note that the group product of these transformations is

$$(\boldsymbol{R}, \boldsymbol{t})(\boldsymbol{O}, \boldsymbol{s}) = (\boldsymbol{R}\boldsymbol{O}, \boldsymbol{R}\boldsymbol{s} + \boldsymbol{t}).$$

We need to show $(\boldsymbol{R}\boldsymbol{O}, \boldsymbol{R}\boldsymbol{s} + \boldsymbol{t}) \in \mathcal{F}((\boldsymbol{R}, \boldsymbol{t})\boldsymbol{X})$. Indeed, $\boldsymbol{R}\boldsymbol{O} \in O(3)$ and consists of eigenvectors of $\boldsymbol{C} = \boldsymbol{R}\boldsymbol{X}^T(\boldsymbol{I} - \frac{1}{n}\mathbf{1}\mathbf{1}^T)\boldsymbol{X}\boldsymbol{R}^T$ as can be verified with a direct computation. If $\boldsymbol{O}, \boldsymbol{R} \in SO(d)$ then also $\boldsymbol{R}\boldsymbol{O} \in SO(d)$. Furthermore

$$\frac{1}{n}\left(\boldsymbol{X}\boldsymbol{R}^T + \mathbf{1}\boldsymbol{t}^T\right)^T \mathbf{1} = \frac{1}{n}\left(\boldsymbol{R}\boldsymbol{X}^T + \boldsymbol{t}\mathbf{1}^T\right)\mathbf{1} = \boldsymbol{R}\boldsymbol{s} + \boldsymbol{t}$$

as required. We have shown $(\boldsymbol{R}, \boldsymbol{t})\mathcal{F}(\boldsymbol{X}) \subset \mathcal{F}((\boldsymbol{R}, \boldsymbol{t})\boldsymbol{X})$ for all $\boldsymbol{X}$ and $(\boldsymbol{R}, \boldsymbol{t})$. To show the other inclusion let $\boldsymbol{X} = (\boldsymbol{R}, \boldsymbol{t})^{-1}\boldsymbol{Y}$ and get $\mathcal{F}((\boldsymbol{R}, \boldsymbol{t})^{-1}\boldsymbol{Y}) \subset (\boldsymbol{R}, \boldsymbol{t})^{-1}\mathcal{F}(\boldsymbol{Y})$ that also holds for all $\boldsymbol{Y}$ and $(\boldsymbol{R}, \boldsymbol{t})$. In particular $(\boldsymbol{R}, \boldsymbol{t})\mathcal{F}(\boldsymbol{X}) \supset \mathcal{F}((\boldsymbol{R}, \boldsymbol{t})\boldsymbol{X})$. The frame $\mathcal{F}$ is bounded since for compact $K \subset \mathbb{R}^{n\times d}$, the translations $n^{-1}\boldsymbol{X}^T\mathbf{1}$ are compact and therefore uniformly bounded for $X \in K$, and orthogonal matrices always satisfy $\|\boldsymbol{R}\|_2 = 1$. $\square$

## A.7 PROOF OF PROPOSITION 2

*Proof.* First note that by definition $S(\mathbf{X})$ is equivariant in rows, namely $S(\rho_1(g)\mathbf{X}) = \rho_1(g)S(\mathbf{X})$. Therefore if $g \in \mathcal{F}(\mathbf{X}) \subset S_n$, then by definition of the frame $\rho_1(g)S$ is sorted. Therefore, $\rho_1(g)S = \rho_1(gh)\rho_1(h)^{-1}S(\mathbf{X}) = \rho_1(gh)S(\rho_1(h)^{-1}\mathbf{X})$ is sorted and we get that $gh \in \mathcal{F}(\rho_1(h)^{-1}\mathbf{X})$. We proved $\mathcal{F}(\mathbf{X})h \subset \mathcal{F}(\rho_1(h)^{-1}\mathbf{X})$ for all $h \in G$ and all $\mathbf{X} \in V$. Taking $\mathbf{X} = \rho_1(h)\mathbf{Y}$ for an arbitrary $\mathbf{Y} \in V$, and $h = g^{-1}$ for arbitrary $g \in G$, we get $\mathcal{F}(\rho_1(g^{-1})\mathbf{Y}) \subset \mathcal{F}(\mathbf{Y})g$, for all $g \in G$ and $\mathbf{Y} \in V$. We proved $\mathcal{F}(\mathbf{X})h = \mathcal{F}(\rho_1(h)^{-1}\mathbf{X})$, which amounts to right action equivariance, see equation 4. The frame is bounded since $\rho_1(G)$ is a finite set. $\qquad\square$

## B  EMPIRICAL FRAME ANALYSIS

**Repeating eigenvalues.** We empirically test the likeliness of repeating eigenvalues in the covariance matrix used from the definition of frame in Section 3.1. We use the data from the $n$-body dataset (Satorras et al., 2021). Let $\mathbf{X} \in \mathbb{R}^{n \times 3}$ represent the set of particles locations (each set centered around $0$ and scaled to have $\max_{x_i \in \mathbf{X}} \|x_i\|_2 = 1$) and $\lambda_1 \le \lambda_2 \le \lambda_3$ be the eigenvalues, sorted in increasing order, of the covariance matrix $C = (\mathbf{X} - \mathbf{1}t^T)^T(\mathbf{X} - \mathbf{1}t^T)$ . In order to measure the proximity of eigenvalues across the dataset we use the notion of eigenvalues spacing $s_i = \frac{\lambda_{i+1} - \lambda_i}{\bar{s}}$, $i = 1, 2$, where $\bar{s} = \frac{\lambda_3 - \lambda_1}{2}$ is the mean spacing. Furthermore we define $s_{min} = \min\{s_1, s_2\}$ as the minimal normalized spacing, a ratio that indicates how close the spectrum is to having repeating eigenvalues. Figure 3 presents a histogram of the minimal spacing over the training set of the $n$-body dataset (consists of 3000 particles sets). The minimal spacing encountered in this experiment is of order $10^{-2}$. This empirically justifies the usage of finite frames for $E(d)$ equivariance.

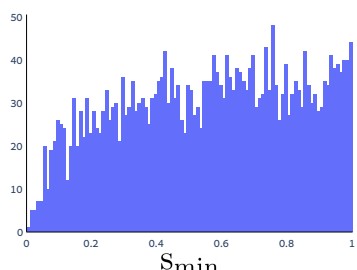

Figure 3: Minimal spacing histogram over the training set of $n$-body dataset (Satorras et al., 2021) .

**Frame stability.** Here we test stability of our $O(d)$ frame defined in Section 3.1. By stability we mean the magnitude of change of a frame w.r.t. the change in the input $\mathbf{X}$. A desired attribute of the constructed frame is to be stable, that is, to exhibit small changes if the input is perturbed. We quantify this stability by comparing the distance between our frames to frames constructed by a noisy input. As in the previous experiment used the data from the $n$-body dataset (Satorras et al., 2021). Let $\mathbf{X} \in \mathbb{R}^{n \times 3}$ represent the set of particles locations (each set centered around $0$ and scaled to have $\max_{x_i \in \mathbf{X}} \|x_i\|_2 = 1$) and $\mathbf{X}_\sigma = \mathbf{X} + \mathbf{Z}$ where $z_{i,j} \sim \mathcal{N}(0, \sigma)$ is the noisy input sample . We compute $\mathcal{F}(\mathbf{X})$ and $\mathcal{F}(\mathbf{X}_\sigma)$ and choose representatives from each set $g = (\mathbf{R}, t) \in \mathcal{F}(\mathbf{X}), g_\sigma = (\mathbf{R}_\sigma, t_\sigma) \in \mathcal{F}(\mathbf{X}_\sigma)$. We measure the distance between the frames as a function of the representatives -

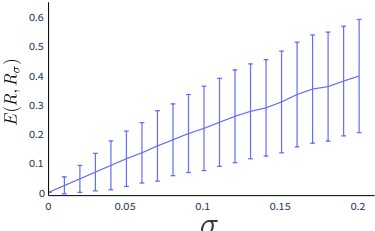

Figure 4: Distance between original and noisy frames as a function of $\sigma$ (we plot average and std). The result is reported over the training set of $n$-body dataset (Satorras et al., 2021) .

$$E(\mathbf{R}, \mathbf{R}_\sigma) = \frac{1}{3} \sum_{i=1}^{3} \sqrt{1 - \langle \mathbf{R}_{:,i}, (\mathbf{R}_\sigma)_{:,i} \rangle^2}$$

Notice that in the case of simple spectrum the distance is invariant to the selection of representatives. In Figure 4 we plot distance of original and noisy frames (and its standard deviation) as function of noise level $\sigma$. The plot validates the continuity of frames in the simple spectrum case.

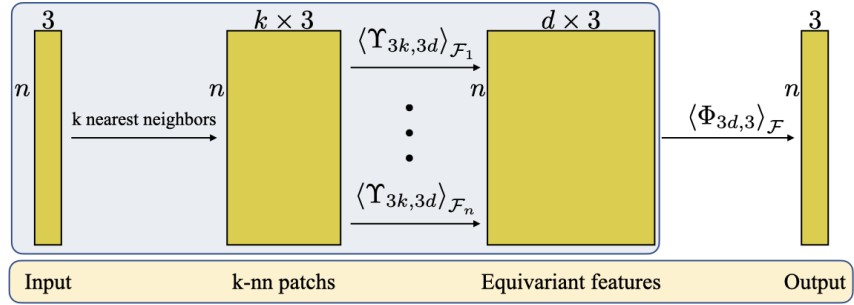

Figure 5: FA-Local-PointNet architecture.

## C  IMPLEMENTATION DETAILS

### C.1  POINT CLOUDS: NORMAL ESTIMATION

Here we provide implementation details for the experiment in section 5.1.

**PointNet architecture.**  Our backbone PointNet is based on the object part segmentation network from Qi et al. (2017a). The network consists of layers of the form

$$\mathrm{FC}(n, d_{\mathrm{in}}, d_{\mathrm{out}}) : \boldsymbol{X} \mapsto \nu\left(\boldsymbol{X}\boldsymbol{W} + \boldsymbol{1}\boldsymbol{b}^T\right)$$
$$\mathrm{MaxPool}(n, d_{\mathrm{in}}) : \boldsymbol{X} \mapsto \boldsymbol{1}[\max \boldsymbol{X}\boldsymbol{e}_i]$$

where $\boldsymbol{X} \in \mathbb{R}^{n \times d_{\mathrm{in}}}$, $\boldsymbol{W} \in \mathbb{R}^{d_{\mathrm{in}} \times d_{\mathrm{out}}}$, $\boldsymbol{b} \in \mathbb{R}^{d_{\mathrm{out}}}$ are the learnable parameters, $\boldsymbol{1} \in \mathbb{R}^n$ is the vector of all ones, $[\cdot]$ is the concatenation operator, $\boldsymbol{e}_i$ is the standard basis in $\mathbb{R}^{d_{\mathrm{in}}}$, and $\nu$ is the ReLU activation. In this experiement, for the PointNet baseline and for $\Phi_{3,3}$ (the backbone in FA-PointNet), we used the following architecture:

$$\mathrm{FC}(512, 3, 64) \overset{L_1}{\to} \mathrm{FC}(512, 64, 128) \overset{L_2}{\to} \mathrm{FC}(512, 128, 128) \overset{L_3}{\to} \mathrm{FC}(512, 128, 512) \overset{L_4}{\to}$$

$$\mathrm{FC}(512, 512, 2048) \overset{L_5}{\to} \mathrm{MaxPool}(512, 2048) \overset{L_6}{\to} [L_1, L_2, L_3, L_4, L_5, L_6] \overset{L_7}{\to}$$

$$\mathrm{FC}(512, 4028, 256) \overset{L_8}{\to} \mathrm{FC}(512, 256, 128) \overset{L_9}{\to} \mathrm{FC}(512, 128, 3).$$

Note that the original PointNet network also contains two T-Net networks, applied to the input and to $L_3$ (the output of the third layer). Similarly, our baseline implementation made a use of the same T-Net networks. Note that the T-Net networks were *not* part of our FA-PointNet backbone architectures $\Phi_{3,3}$.

The FA-Local-Pointnet architecture can be seen as a composition of two parts (see Figure 5). The first part outputs equivariant features by applying the same MLP backbone $\Upsilon_{3k,3d}$ with FA on each point's k-nn patch. Then, from each point's equivariant features, the second part of the network outputs an equivariant normal estimation. Note that the second part, $\langle\Phi\rangle_{\mathcal{F}}$, is an FA-Pointnet network applied to a point cloud of dimensions $\mathbb{R}^{n \times 3d}$. For FA-Local-PointNet we made the following design choices. Each point patch is constructed as its $k$ nearest neighbors, with $k = 20$, $\boldsymbol{X}_i \in \mathbb{R}^{20 \times 3}$. Then, the backbone $\Upsilon_{3*20,3*42}$ is applied to all patches, with the following PointNet layers:

$$\mathrm{FC}(512, 3*20, 3*21) \to \mathrm{FC}(512, 3*21, 3*42) \to \mathrm{FC}(512, 3*42, 3*42).$$

The second backbone, $\Phi_{3*42,3}$, is built from the following PointNet layers:

$$\mathrm{FC}(512, 3*32, 128) \overset{L_1}{\to} \mathrm{FC}(512, 128, 256) \overset{L_2}{\to} \mathrm{FC}(512, 256, 256) \overset{L_3}{\to} \mathrm{FC}(512, 256, 512) \overset{L_4}{\to}$$

$$\mathrm{FC}(512, 512, 2048) \overset{L_5}{\to} \mathrm{MaxPool}(512, 2048) \overset{L_6}{\to} [L_1, L_2, L_3, L_4, L_5, L_6] \overset{L_7}{\to}$$

$$\mathrm{FC}(512, 5248, 256) \overset{L_8}{\to} \mathrm{FC}(512, 256, 128) \overset{L_9}{\to} \mathrm{FC}(512, 128, 3).$$

**DGCNN architecture.** Our backbone DGCNN architecture, $\Phi_{3,3}$, is based on the object part segmentation network from Wang et al. (2018). It consists of $\text{EdgeConv}(n, d_{\text{in}}, d_{\text{out}})$, $\text{FC}(n, d_{\text{in}}, d_{\text{out}})$, and $\text{MaxPool}$ layers.

$$\text{EdgeConv}(512, 3, 64) \overset{L_1}{\to} \text{EdgeConv}(512, 64, 64) \overset{L_2}{\to} \text{EdgeConv}(512, 64, 64) \overset{L_3}{\to}$$

$$\text{FC}(512, 64, 1024) \overset{L_4}{\to} \text{MaxPool}(512, 1024) \overset{L_5}{\to} [L_1, L_2, L_3, L_4, L_5] \overset{L_6}{\to}$$

$$\text{FC}(512, 1216, 256) \overset{L_7}{\to} \text{FC}(512, 256, 128) \overset{L_8}{\to} \text{FC}(512, 128, 3).$$

Note that the DGCNN architecture incorporates T-Net network applied to the input.

**Training details.** We trained our networks using the ADAM (Kingma & Ba, 2014) optimizer, setting the batch size to 32 and 16 for PointNet and DGCNN respectively. We set a fixed learning rate of 0.001. All models were trained for 250 epochs. Training was done on a single Nvidia V-100 GPU, using PYTORCH deep learning framework (Paszke et al., 2019).

## C.2 GRAPHS: EXPRESSIVE POWER

We provide implementation details for the experiments in 5.2. We used two different universal backbones, MLP and GIN equipped with identifiers. The details of those architectures are presented here.

**FA/GA GIN+ID architecture.** The GIN+ID backbone is based on the GIN (Xu et al., 2018a) network with an addition of identifiers as node features in order to increase expressiveness. For the experiments we used a three-layer GIN with a feature dimension of size 64 and a ReLU activation function. For the added identifiers we defined the input node features as $\boldsymbol{X}^0 = [\boldsymbol{X}^0, \boldsymbol{I}_n]$, where $n$ is the number of nodes in the graph, and to handle graphs of different sizes (EXP dataset) we padded the node features with zeros to fit the size of the maximal graph. Note that we did not apply the permutation generated by the frame on the identifiers.

**FA/GA MLP architecture.** We used two different MLP networks for the EXP dataset and the GRAPH8c, due to the different size of graphs in the datasets. In the EXP dataset the maximal graph size is 64 and every node in the graph has a one dimensional binary feature, therefore the input for the MLP network is a flatten representation of the graph (with additional padding according to the graph size) $\boldsymbol{x} \in \mathbb{R}^{64^2+64}$. Our architecture consists of layers of the form

$$\text{FC}(d_{\text{in}}, d_{\text{out}}) : \boldsymbol{x} \mapsto \nu(\boldsymbol{W}\boldsymbol{x} + \boldsymbol{b})$$

where $\boldsymbol{W} \in \mathbb{R}^{d_{\text{out}} \times d_{\text{in}}}$, $\boldsymbol{b} \in \mathbb{R}^{d_{\text{out}}}$ are the learnable parameters. The final output of the network is denoted by $(d_{out})$ where $d_{out}$ is the output dimension.

The MLP network structure for the EXP-classify task:

$$\text{FC}(4160, 2048) \overset{L_1}{\to} \text{FC}(2048, 4096) \overset{L_2}{\to} \text{FC}(4096, 2048) \overset{L_3}{\to}$$

$$\text{FC}(2048, 10) \overset{L_4}{\to} \text{FC}(10, 1) \overset{L_5}{\to} (1)$$

with ReLU as the activation function. For the EXP task, which has 10-dimensional output for each graph we omitted the last layer. The GRAPH8c is composed of all the non-isomorphic connected graphs of size 8, hence we did not used any padding of the input here. The nodes have no features and we just used a flatten version of the adjacency matrix. The architecture for the GRAPH8c task:

$$\text{FC}(64, 128) \overset{L_1}{\to} \text{FC}(128, 64) \overset{L_2}{\to} \text{FC}(64, 10) \overset{L_3}{\to} (10)$$

**Training details.** We followed the protocol from (Balcilar et al., 2021) and trained our model with batch size 100 for 200 epochs. The learning rate was set to 0.001 and did not change during training. For optimization we used the ADAM optimizer. Training was done on a single Nvidia RTX-8000 GPU, using PYTORCH deep learning framework.

**Invariance Evaluation.**   We quantify the permutation invariance of a model $\phi$ by comparing output of randomly permuted graphs $\rho_1(g_i)\mathbf{X}$, $g_i \in S_n$, $i \in [50]$. The evaluation metric used is the *invariance error*, defined by

$$\frac{1}{m} \sum_{i=1}^{m} \|\phi(\rho_1(g_i)\mathbf{X}) - \boldsymbol{v}\|_2\,,$$

where $\boldsymbol{v} = \frac{1}{m}\sum_{i=1}^{m} \phi(\rho_1(g_i)\mathbf{X})$.

The permutation *invariance error* of the models (FA/GA) was measured as a function of the sample size $k$. We iterated over the entire GRAPH8c dataset and for every graph computed the *invariance error* for the FA-MLP ,GA-MLP and regular MLP models (all with the exact same backbone network). The results presented in Figure 2 (left) are normalized by the error of the regular MLP model. The backbone MLP we used for this experiment is of the form:

$$\text{FC}(64, 128) \overset{L_1}{\to} \text{FC}(128, 128) \overset{L_2}{\to} \text{FC}(128, 128) \overset{L_3}{\to} \text{FC}(128, 10) \overset{L_4}{\to} (10)$$

## C.3   GRAPHS: $n$-BODY PROBLEM

This section describes the implementation details for the $n$-body experiment from Section 5.3.

**FA-GNN architecture.**   We use the GNN architecture of Gilmer et al. (2017) our base GNN layer, where for each node $i \in [n]$ the update rule follows,

$$\boldsymbol{m}_{ij} = \phi_e(\boldsymbol{h}_i^l, \boldsymbol{h}_j^l, a_{ij})$$

$$\boldsymbol{m}_i = \sum_{j\in\mathcal{N}(i)} \boldsymbol{m}_{ij}$$

$$\boldsymbol{h}_i^{l+1} = \phi_h(\boldsymbol{h}_i^l, \boldsymbol{m}_i)$$

where $\mathcal{N}(i)$ are the indices of neighbors of vertex $i$, $\boldsymbol{h}_i^l$ is the embedding of node $i$ at layer $l$ and $a_{ij} \in \{-1, 1\} \times \mathbb{R}^+$ are edge attributes representing attraction or repelling between pairs of particles and their distance. $\boldsymbol{h}^0 \in \mathbb{R}^{n\times 6}$ represents the nodes input features, which in this experiment are a concatenation of the nodes initial 3D position (rotation and translation equivariant) and velocities (rotation-equivariant and translation-invariant). We used node feature dimension of size 60 to maintain a fair comparison with the baselines (Satorras et al., 2021). The functions $\phi_e$ and $\phi_h$ are implemented as a two-layer MLP with the SiLU activation function (also chosen for consistency purposes) with an hidden dimension of 121 and 120 respectively. To maintain fair comparison with (Satorras et al., 2021) our network is composed of 4 GNN layers. $\Phi_{6,3d'}^{(1)}$ has an additional linear embedding of the features as a prefix to the GNN layers while $\Phi_{3d',3}^{(4)}$ is equipped with a two-layer MLP (SiLU activation) as a decoder to extract final positions.

**Training details.**   We followed the protocol from (Satorras et al., 2021) and trained our model with batch size 100 for 10000 epochs. The learning rate was set to 0.001 and did not changed during training. For optimization we used the ADAM optimizer. Training was done on a single Nvidia RTX-6000 GPU, using PYTORCH deep learning framework.

