# OpenReview forum: "Frame Averaging for Invariant and Equivariant Network Design"
_ICLR.cc/2022/Conference — ICLR 2022 Oral_

### Official Review · Reviewer_5HAS · 2021-11-02

**Correctness:** 3
**Technical Novelty And Significance:** 4
**Empirical Novelty And Significance:** 3
**Recommendation:** 8
**Confidence:** 2

**Main Review:**

*Strengths*
- The idea of replacing the averaging operator over the entire group by the FA is innovative. FA is simpler to compute and has less complexity in comparison with the averaging operator over the entire group.

- The authors prove that the FA-based models preserve the universality property of their backbone architectures.

- The FA framework is then applied to design several invariant and equivariant architectures for point clouds and graphs.

*Weaknesses*

- My concern is mostly about the incompleteness of the framework. It may happen the case that for some $\mathbf{X}$, $\mathcal{F}(\mathbf{X})$ is a small set, but for some other $\mathbf{X}$, $\mathcal{F}(\mathbf{X})$ is large or even infinity. (For example the frame choice in Subsection 3.1 is in this case). In this case, a deeper analysis on how to separate these two cases and how to deal when $\mathcal{F}(\mathbf{X})$ is large or even infinity is necessary.

- It is also not clear how the FA is adapted in the DA-Local-PointNet. A detailed explanation here would be helpful.

- No comparison with the previous equivariant architectures are presented. Therefore, it is hard to estimate how novel and efficient the framework is in the world of current group equivariant architectures.

- Some technical parts are quite compact and not easy to read. Maybe the reason is that the paper includes rich contents and the authors tried to fix all of them in 9 pages.

**Summary Of The Paper:**

The authors introduce Frame Averaging (FA), a general framework for adapting known architectures to become invariant or equivariant with respect to a general group by using group averaging operator. The idea of FA is to replace the averaging operator over the entire group by the averaging over a smaller set of group elements but still achieve the full invariant/equivariant property.

**Summary Of The Review:**

In addition, I have some comments on the technical parts of the paper:

- The definition of the frame $\mathcal{F}(\mathbf{X})$ for the case of point clouds and $G=E(d)$ in page 5 is not clear. Why "$\mathcal{F}(\mathbf{X})$ [...] is the collection of E(3) Euclidean transformations defined by [...]", while in this case, I think, $\mathcal{F}(\mathbf{X})$ must be a subset of $E(d)$?

- I am curious to know why "generically the frame would consist of $2^d$ elements [...], while for rare inputs $\mathbf{X}$ the frame can be an infinite set"? I think this remark is crucial for the proposed method as it affects the size of the FA framework.

- I think the proof of Proposition 1 (Appendix A.6) has some flaws:

     -- Line 3: What is $\Lambda$? It is not defined yet.

     -- Line 6-7: It seems to me that, you need to prove that $\mathbf{RO}$ consists of eigenvectors of
$$[(\mathbf{R},\mathbf{t})\mathbf{X}]^T \cdot (I-\frac{1}{n} \mathbf{1} \mathbf{1}^T) \cdot [(\mathbf{R},\mathbf{t})\mathbf{X}]$$
which is
$$(\mathbf{R}+\mathbf{t} \mathbf{1}^T) \mathbf{X}^T \cdot (I-\frac{1}{n} \mathbf{1} \mathbf{1}^T) \cdot \mathbf{X} (\mathbf{R}+\mathbf{t} \mathbf{1}^T)^T$$
rather than
$$C = \mathbf{R} \mathbf{X}^T \cdot (I-\frac{1}{n} \mathbf{1} \mathbf{1}^T) \cdot \mathbf{X} \mathbf{R}^T$$
as you claimed. If this is exactly the case, I do not know how it can be true.

---

> ### Author Response · Authors · 2021-11-19
> **Official Response to Reviewer 5HAS**
>
> Thank you for your detailed review. Below we address the main comments and
> questions expressed in this review. All changes refer to the uploaded revision.
>
> **Q:** My concern is mostly about the incompleteness of the framework. It may happen that for some $X$, $F(X)$  is a small set, but for some other $X$, $F(X)$  is large or even infinity. In this case, a deeper analysis on how to separate these two cases is necessary. I am curious to know why "generically the frame would consist of $2^d$  elements [...], while for rare inputs $X$  the frame can be an infinite set"?
> **A:** Thank you for this comment. First, although the infinite frame case if of real interest this paper deals exclusively with **finite** frames. We made that clear in the revision: We added a clear statement at Section 2.1 after Theorem 1, and in Section 2.2 dealing with expressive power, we explicitly state we work in domains where the frame is finite. Furthermore, we have rewritten two paragraphs in Section 3.1 explicitly defining the frame for simple spectrum covariance matrices. We hope it is clearer now why it has $2^d$ elements in this case, and also provide a citation for the genericity claim [1]. In Appendix B (“repeating eigenvalue”) we provide an empirical validation to the genericity claim, namely that all covariance matrices that we encounter have a simple spectrum. Note that in this case the frame size is **constant** where it is defined. In the case of discrete groups (i.e., permutations) there is no natural notion of continuity and indeed frame sizes can change. This seems necessary in this case since, for example, different input graphs could have different self symmetries (automorphisms).
>
> [1] On the geometry of the set of symmetric matrices with repeated eigenvalues,
> Breiding, Kozhasov, Lerario, Arnold Mathematical Journal 4(3), 2018, Springer.
>
> **Q:** It is also not clear how the FA is adapted in the FA-Local-PointNet. A detailed explanation here would be helpful.
> **A:** Thank you for this comment. We added more details about the FA-Local-Pointnet construction in the appendix, including a visualisation. Please see the appendix, section C.1.
>
>
> **Q:** No comparison with the previous equivariant architectures are presented.
> **A:** This is actually not true. We have compared to **4** state-of-the-art equivariant architectures. In more detail:
> In 5.1 we compared to the Vector Neurons model [1].
> In 5.2 we compared to SE(3) Transformer [2], TFN [3], and EGNN [4].
> We strongly believe that this collection of previous works is a solid representation of
> current group equivariant architectures.
>
> [1] Vector neurons: A general framework for so (3)-equivariant networks. Deng et al, ICCV 2021.
> [2] SE(3)-Transformers: 3D Roto-Translation Equivariant Attention Networks, Fuchs et al, 34th Conference on Neural Information Processing Systems (NeurIPS 2020).
> [3]  Tensor field networks: Rotation- and translation-equivariant neural networks for 3D  point clouds, Thomas et al, 32th Conference on Neural Information Processing Systems (NeurIPS 2018).
> [4] E(n) Equivariant Graph Neural Networks, Satorras et al, International Conference on Machine Learning 2021.
>
> **Q:** The definition of the frame $F(X)$  for the case of point clouds and $G=E(d)$  in page 5 is not clear. Why " $F(X)$  [...] is the collection of $E(3)$ Euclidean transformations defined by [...]", while in this case, I think,  $F(X)$  must be a subset of  $E(d)$ ?
> **A:** We have realized the previous frame definition was unclear and rewritten that part in Section 3.1.
>
>
> **Q:** I think the proof of Proposition 1 (Appendix A.6) has some flaws:
> What is $\Lambda$?
>  (b) you need to prove that $RO$ consists of eigenvectors of $[(R,t)X]^T⋅(I−\frac{1}{n}11^T)⋅[(R,t)X]$.
> which is - $(R+t1^T)X^T⋅(I−\frac{1}{n}11^T)⋅X(R+t1^T)^T$
> rather than - $C=RX^T⋅(I−\frac{1}{n}11^T)⋅XR^T$
> ...I do not know how it can be true.
> **A:** For (a), $\Lambda$ is defined to be the eigenvalues (diagonal) matrix in the eigen-decomposition of $C$ (updated in the revision). For (b), we had a typo in the definition of the group action (corrected in the revision) -
> Old version was $\rho_1(g)X = X(R^T+1t^T)$
> Instead of $\rho_1(g)X = XR^T+1t^T$.
> Which is now fixed and consistent with the proof of proposition 1.

---

> > ### Comment · Reviewer_5HAS · 2021-11-22
> > **The authors have solved the technical issues I raised**
> >
> > Thank you very much for the authors' responses! The responses are quite helpful and clear. The technical issues I raised have been clarified and fixed. Therefore, I increase the score to 8.

---

### Official Review · Reviewer_BhKC · 2021-11-02

**Correctness:** 4
**Technical Novelty And Significance:** 4
**Empirical Novelty And Significance:** 4
**Recommendation:** 8
**Confidence:** 5

**Details Of Ethics Concerns:**

There are no ethical issues in this paper.

**Main Review:**

Strengths
This paper proposes a general method for transforming existing models into invariant/equivariant models using Reynolds operators.
Combined with other methods, it provides state-of-the-art experimental results.

Weaknesses & Questions
First of all, I would like to point out that the idea of using Reynolds operators to transform a model into an invariant model is found in Kicki et al. 2021, where the construction is exactly the same except for the use of frames.
The authors should be cited for this paper.
Also, the name "frame" is confusing with the concept of frame in differential geometry and should be given a different name.

Theorem 1: It is proved that if a frame is an equivariant function, then a partial sum over the frame gives a transformation to an invariant/equivariant function, but the fact that the frame is defined depending only on the input space is not appropriate.
For example, if we have an invariant model F, the proposed method will sum over frames to convert it to an invariant function, which will increase the computational complexity. The transformation should be done without any transformation for the invariant model F.
This leads to the problem of overlapping invariants when combined with other models in the experimental section.

Also, since calculating the specific frame itself involves mathematical difficulties, I believe that the evaluation should be done on the model for which the frame has been calculated, i.e., for which the experiment has been conducted.

Point cloud models:
The $S_n \times E(d)$ or $SE(d)$-invariant model is constructed by computing a frame for $E(d)$ or $SE(d)$ and transforming the model of the Deep sets with that frame.
The simple question is why don't you construct a frame for $S_n \times E(d)$ or $SE(d)$?
If this method is good enough, the point permutation action should also be subject to the frame averaging model.
I would like you to explain the rationale reason for not doing so.

Graph models:
Two types of models have been proposed: MLP+FA and GNN+FA.
The problem with MLP+FA is that it uses the adjacency matrix itself as input, so when the number of nodes is large, the input space is also large, and the number of parameters is significantly larger than, for example, the model in Maron et al.
Is it possible to train MLP+FA with, say, 50 nodes?
Also, in a real task, the number of nodes in a graph can take many different values, is it possible to train MLP+FA for such a case?
In GNN+FA, the problem seems to be that invariance is calculated redundantly as described above, and the contribution of this method is not clear.

Reference

Zaheer et al.
Deep sets
Neural Information Processing Systems (NeurIPS) 2017.

Maron et al. Invariant and Equivariant Graph Networks
International Conference on Learning Representations (ICLR) 2019

Kicky et al.
A New Approach to Design Symmetry Invariant Neural Networks
International Joint Conference on Neural Networks (IJCNN) 2021





**Summary Of The Paper:**

This paper proposes a method to transform a model into an invariant/equivariant model using the Reynolds operator.
In addition, they define a notion called the equivariant frame to compute the Reynolds operator efficiently.

**Summary Of The Review:**

The structure of the proposed model has already been seen in Kicki et al. 2021 except for the use of frames, and even if frames are used, MLP+FA for example does not seem to be able to handle changes in inputs such as changes in the number of nodes in the graph.
When combined with the existing strong methods, it gives good results, but this could be achieved by, for example, concatting two GNNs and then transforming them with FNN, so we decided that it is not worthy of evaluation.

After some discussion, the score was raised because the theoretical uncertainties and doubts were resolved.

---

> ### Author Response · Authors · 2021-11-19
> **Official Response to Reviewer BhKC 1/2**
>
> We want to thank the reviewer for the review. Unfortunately, we feel that the reviewer has misunderstood a core concept of the paper, which we try to clarify below.
>
> **Q:** This paper proposes a method to transform a model into an invariant/equivariant model using the Reynolds operator. In addition, they define a notion called the equivariant frame to compute the Reynolds operator efficiently.
> **A:** Our paper does **not** try to provide efficient computation, nor an approximation to the Reynolds operator (group averaging).  Instead, we define a new operator, the Frame Averaging (FA) operator, that has two crucial differences to the Reynolds operator: (i) FA averages on a **subset** of the group; and (ii) FA is input dependent. The FA operator can yield arbitrary different results than the Reynolds operator that averages over the entire group. As mentioned after Theorem 1, FA is a generalization of group averaging. In fact, we even compare with the Reynolds operator (called GA, acronym for Group Averaging) in Section 5.2, showing superiority of FA in performance and computational complexity.
>
>
> **Q:** I would like to point out that the idea of using Reynolds operators to transform a model into an invariant model is found in Kicki et al. 2021, where the construction is exactly the same except for the use of frames. The authors should be cited for this paper. (Kicky et al. A New Approach to Design Symmetry Invariant Neural Networks International Joint Conference on Neural Networks (IJCNN) 2021).
> **A:** First, let us note that our paper cites **earlier** papers that suggest and use the Reynolds operator than Kicki et al. 2021. This is done before equation 1 in the paper: Murphy ey al. 2018, and Yarotsky who’s original preprint dates back to 2018.  Second, regarding Kicki et al., note that this work considers subgroups of $S_n$, therefore restricts to the finite group case (cannot handle infinite groups such as $E(3)$), and the number of terms in the sum is much larger than required for Frame averaging. In Section 5.2 we compare FA to averaging of $S_n$ using the Reynold operator.
>
> **Q**. The name "frame" is confusing with the concept of frame in differential geometry and should be given a different name.
> **A:** Actually, our frames do relate to frames in differential geometry. Let us try to explain: The method of moving frames, and in particular Darboux frames, are chosen on a surface in $\mathbb{R}^3$ in an $E(3)$ equivariant manner by aligning them with principal directions and the normal to the surface. If the surface rotates and/or translates the Darboux frame changes accordingly. Our frames can be seen as a generalization of this principle.
> Further note that also Darboux frames are defined everywhere on the surface **except** at umbilical points, which are defined by repeating eigenvalues of the shape operator, similar to the situation with our $E(d)$ frame construction.
>
>
> **Q:** Theorem 1: It is proved that if a frame is an equivariant function, then a partial sum over the frame gives a transformation to an invariant/equivariant function, but the fact that the frame is defined depending only on the input space is not appropriate. For example, if we have an invariant model F, the proposed method will sum over frames to convert it to an invariant function, which will increase the computational complexity. The transformation should be done without any transformation for the invariant model F. This leads to the problem of overlapping invariants when combined with other models in the experimental section.
> **A:** We didn’t fully understand this comment. We would like to point out a few aspects that perhaps were misunderstood:
> 1. Our goal in this work is to transform non-invariant/equivariant (backbone) architectures to become invariant or equivariant to new symmetry types. Therefore we **never** apply Frame Averaging to an already invariant/equivariant mode.
> 2. We are not using a partial sum. As mentioned above, our goal is not to compute (or approximate) the full group average but to average over an (equivariant) subset of the group. This operator could have an arbitrarily different value than the full group average.

---

> > ### Author Response · Authors · 2021-11-19
> > **Official Response to Reviewer BhKC 2/2**
> >
> > **Q:** Point cloud models: The $S_n\times E(d)$ or $SE(d)$-invariant model is constructed by computing a frame for $E(d)$ or $SE(d)$  and transforming the model of the Deep sets with that frame. The simple question is why don't you construct a frame for $S_n\times E(d) $ or $ SE(d)$? If this method is good enough, the point permutation action should also be subject to the frame averaging model. I would like you to explain the rationale reason for not doing so.
> > **A:** Although frame averaging is general and can be used to construct invariant/equivariant models to any given group G including $S_n \times E(d)$ it should be used wisely. Frame averaging provides a remedy to the fact that for certain group representations (e.g., for graph functions, and to an extent, point clouds), it is not clear how to construct efficient architectures which are both expressive (i.e., universal) and invariant/equivariant. However, it has a cost: it does not use parameter sharing and requires more function evaluations (as a function of the sizes of the frames). In the specific case of sets, efficient universal invariant/equivariant architectures do exist, i.e., DeepSets. It is therefore redundant to apply frame averaging over the permutation representation for sets. Using Frame Averaging to **add** additional symmetry (e.g., to build a universal E(3) invariant/equivariant point cloud network) is therefore a virtue.
> >
> >
> >
> > **Q:** The problem with MLP+FA is that it uses the adjacency matrix itself as input, so when the number of nodes is large, the input space is also large, and the number of parameters is significantly larger than, for example, the model in Maron et al. Is it possible to train MLP+FA with, say, 50 nodes?
> > **A:** First, Maron et al. model is known to be not universal [1]. To make it universal, [2] shows a construction that requires high dimensional tensors, presenting a worse (i.e., exponential) tradeoff between expressiveness and complexity than Frame Averaging. To the best of our knowledge the best expressiveness achieved so far with a quadratic model (i.e., using the adjacency as input) is 3-WL [3], which is also not universal. In fact, as far as we can tell, FA provides the **first** quadratic universal model for graphs. Second, the EXP dataset in Section 5.2 contains graphs with up to 64 nodes, larger than 50, and therefore it is indeed feasible to train models on graphs of that size.
> >
> > [1] Expressive Power of Invariant and Equivariant Graph Neural Networks,  Azizian et al., International Conference on Learning Representations (ICLR) 2021
> > [2] On the Universality of Invariant Networks,  Maron et al., International Conference on  Machine Learning (ICML) 2019
> > [3] Provably Powerful Graph Networks,  Maron et al., Conference on Neural Information Processing Systems (NeurIPS 2019)
> >
> > **Q:** In a real task, the number of nodes in a graph can take many different values, is it possible to train MLP+FA for such a case?
> > **A:** The solution for such a case is to pad the input according to the size of the largest graph in the dataset. In the EXP dataset (Section 5.2) the sizes of the graphs change from 32 and to 64 and we use padding to process input for the MLP+FA model.  (This is mentioned in appendix C.2.)
> >
> > **Q:** In GNN+FA, the problem seems to be that invariance is calculated redundantly as described above, and the contribution of this method is not clear.
> > **A:** We respectfully disagree. There are two variants of our implementation to GNN+FA and in both of them the FA procedure is **not** redundant:
> > 1. In 5.2 the addressed symmetry is E(3) (the particles’ positions and velocities as node features), which GNN is not equivariant to, and therefore the frame averaging is required.
> > 2.  In 5.3 we used the GNN equipped with node identifiers (GIN+ID), an universal but not permutation equivariant model. The additional identifiers ruin the equivariance properties of GNN and in order to obtain an equivariant model FA is necessary.
> >
> > **Q:** it gives good results, but this could be achieved by, for example, concatting two GNNs and then transforming them with FNN, so we decided that it is not worthy of evaluation.
> > **A:** We are not clear what was the intention here and we would be thankful if the reviewer could clarify.
> >
> > **Q:** Since calculating the specific frame itself involves mathematical difficulties, I believe that the evaluation should be done on the model for which the frame has been calculated, i.e., for which the experiment has been conducted.
> > **A:** We didn’t understand this comment and would be thankful if the reviewer can rephrase so we can properly respond.

---

> > > ### Comment · Reviewer_BhKC · 2021-11-23
> > > **Thank you for your reply.**
> > >
> > > Thank you for your reply. Many of my questions have been answered.
> > >
> > > My main concern is that because the Frame is tied to an action on the input space, you may be missing cases where a smaller Frame is possible depending on the model.
> > > For example, let $f$ be an $A_n$-invariant function,
> > > then Reynolds operator can be computed over two elements, id and (12).
> > > In this way, when the model is fixed, it is possible to get smaller frames.
> > >
> > > This approach has its merits in that it can be adapted to all models, but I would like you to write a Question in your paper so that the community is aware of this point.
> > > Question: Can model-dependent Frames be found?
> > >
> > > After checking this part, I am thinking of raising my score too.

---

> > > > ### Author Response · Authors · 2021-11-23
> > > > **Interesting observation, suggested change**
> > > >
> > > > Thank you for your response and for acknowledging the merits of the suggested approach.
> > > > Regarding your concern: If we understand correctly, the example you are describing can be formulated in the following general terms: consider a backbone model $\phi$ that is invariant to $H$, which is a normal subgroup of $G$, namely, $H\triangleleft G$. Then, making $\phi$ invariant to $G$ can be done by averaging over arbitrarily chosen representatives from the quotient $G/H$. In some cases, e.g., $A_n \triangleleft S_n$ such quotient is small and full group averaging is tractable. We agree and think this is an interesting observation, thank you!
> > > > Note that in some other cases the quotient group can still be too large for full averaging and could benefit from FA, as well.
> > > > To address your request, we suggest the following addition to the conclusion/future work part of the paper:
> > > > “Another interesting open question for function symmetrization is extending the invariance (equivariance) of a model $\phi$ ($\Phi$) from a subgroup $H$ of $G$, namely $H<G$, to $G$.”
> > > > **Please note:** Since replacing the revision PDF is no longer possible in the system at this time, we will incorporate such a sentence in the next revision cycle.

---

> > > > > ### Comment · Reviewer_BhKC · 2021-11-24
> > > > > **Thank you for your response.**
> > > > >
> > > > > Thank you for your reply. First of all, your generalization is correct (It is not necessary that $H$ be normal).
> > > > >
> > > > > However, I have some doubts about the proposed sentence. This is because, in general, the set of representatives does not have a group structure, as in the case where $H$ is not a normal subgroup.
> > > > >
> > > > > Therefore, the proposed sentence should be "Another interesting open question for function symmetrization is extending the invariance (equivariance) of a model $ \phi (\Phi)$ from a minimal subset $H_{\phi (\Phi)}$ of $G$."
> > > > >
> > > > > Please let me know what you think.

---

> > > > > > ### Author Response · Authors · 2021-11-24
> > > > > > **Clarifying the non-normal case, and another suggestion for additional sentences**
> > > > > >
> > > > > > We agree that averaging over representatives from **all** cosets lead to an efficient computation of the group averaging (Reynolds operator), and hence to an invariant function. However, the number of cosets can be large (or even infinite), and our point is that we are not sure there exists a **subset** of the representatives, which are not input dependent like a frame, that leads to an invariant function in all cases.
> > > > > > We also want to make the point that efficient computation of the Reynolds operator (when it is possible) is a different approach to invariance than Frame Averaging and deserves an independent treatment. If the reviewer agrees we therefore suggest the following *two* sentences:
> > > > > >  “Another interesting open question for function symmetrization is extending the invariance (or equivariance) of a model $\phi$ (or $\Phi$) from a (not necessary normal) subgroup $H$ of $G$, to $G$. A different approach for building efficient invariant/equivariant operators is to come up with an efficient computation to the full group averaging operator.”

---

> > > > > > > ### Comment · Reviewer_BhKC · 2021-11-28
> > > > > > > **Very satisfied.**
> > > > > > >
> > > > > > > Sorry for the late reply.
> > > > > > > I think the proposed text has been improved.
> > > > > > >
> > > > > > > I believe that your model of the point cloud is successful in adding invariance efficiently because the group is decomposed into a direct product.
> > > > > > > However, as the $A_n$ example above shows, I thought that the frame-like set should change depending on the base model as well as the input.
> > > > > > > In any case, I think we had a good discussion.
> > > > > > >
> > > > > > > I change my score to 8.

---

### Official Review · Reviewer_57mq · 2021-11-02

**Correctness:** 4
**Technical Novelty And Significance:** 4
**Empirical Novelty And Significance:** 4
**Recommendation:** 8
**Confidence:** 4

**Details Of Ethics Concerns:**

No ethics concerns

**Main Review:**

**Strengths**:

- *FA is a very simple framework yet it is potentially very useful*. Group equivariance is an important form of inductive bias in deep learning architectures, but designing architectures that has such equivariance is challenging. It would be very useful if we could adapt any back architecture to become invariant/equivariant to a certain group. However, the previously studied group averaging is computationally infeasible when the group is large, so this paper, which greatly reduces the computational cost of group averaging, could be potentially very useful. I really like this work, and will be looking forward to seeing future development of this work.
- *Technical statements are all sound and proved*. Mathematical statements in this paper all look correct to me, and they have provided proofs. These statements are clear and rigorous.
- *Impressive results*. Using the simple idea of frame averaging, the paper demonstrates state-of-the-art results on several tasks. The results are impressive, which suggests that the FA framework can be very useful in practice.

**Weaknesses**:
- *Lack of simple examples*. While it is not hard to check the correctness of all these statements, it takes me some time to form an intuition of what is proposed in this paper. It would help me a lot if the authors could provide a simple example at the beginning to give readers some intuition. For example, it might be good to work through an example to make MLP translation equivariant (with the simplest possible construction of frames)?
- *Insufficient study and explanation of the proposed frames*.
  - The construction of frames in Section 3.1 seems to come out of nowhere. While I could check they are indeed equivariant, I don’t think I understand the motivation or thinking process behind such designs.
  - Furthermore, can the frames be simplified? As an example, if we let the input $X$ be a function on groups, i.e. $X=f(g),g\in G$. Can we let $F(f) = \arg\max_{g} ||f(g)||$? I might be wrong, but I think this simple construction is also equivariant?
  - Are the number of elements output by frames the smaller the better, or is there a balance between performance and computational efficiency?

**Questions**:
*How stable are the proposed frames*? For the proposed frames, I would be interested to know how stable they are? That is to say, if I add noise to the inputs, will the output subset of groups be significantly different?


**Summary Of The Paper:**

**Summary and Contributions**:

The paper introduces a framework called Frame Averaging (FA) that can adapt existing backbone architectures to become invariant/equivariant to new symmetry types. It achieves this by averaging over an input-dependent frame which outputs a subset of groups. Frame averaging is often much more efficient to compute than averaging over the entire group, while at the same time, guarantees exact invariance/equivariance.

On the technical side, the paper also proves that FA-based models have the same expressive power as the original backbone architectures.

On the empirical side, the paper provides new classes of models using FA such as universal Euclidean motion invariant point cloud networks / Message Passing GNNs, and demonstrates their practical effectiveness on several tasks.


**Summary Of The Review:**

The paper studies an important problem that could potentially have great impact: How to adapt existing architectures to become invariant/equivariant to a certain group while maintaining the expressive power and computational efficiency of the original backbone model? The paper provides a simple yet effective solution. The technical statements are sound and the empirical results are impressive.

---

> ### Author Response · Authors · 2021-11-19
> **Official Response to Reviewer 57mq**
>
> Thank you for your detailed review. Below we address the main comments and
> questions expressed in this review. All changes refer to the uploaded revision.
>
> **Q:** Simple examples. It takes me some time to form an intuition of what is proposed in this paper. It would help me a lot if the authors could provide a simple example at the beginning to give readers some intuition. For example, it might be good to work through an example to make MLP translation equivariant?
> **A:** Added a simple running example in Section 2.1.
>
> **Q:** The construction of frames in Section 3.1 seems to come out of nowhere. While I could check they are indeed equivariant, I don’t think I understand the motivation or thinking process behind such designs.
> **A:** The role of frames is to normalize the data so that the network always “sees” symmetric versions of the input in the same way. This is why choosing a frame requires finding a property of the data that is equivariant and using that to define the frame. One intuition comes from differential geometry where moving frames, and in particular Darboux frames are set on a surface by aligning with principal directions and the surface’s normal. If the surface rotates and/or translates the Darboux frame changes accordingly. Our frames can be seen as a generalization of this principle.
>
>
> **Q:** Can the frames be simplified? As an example, if we let the input $X$  be a function on groups, i.e. $X=f(g), g∈G$ . Can we let $F(f)=\arg⁡max_g||f(g)||$ ? I might be wrong, but I think this simple construction is also equivariant?
> **A:** The reviewer is right, the frame suggested is indeed equivariant. As the reviewer witnessed, frame design is a very flexible framework that could be constructed with different approaches, optimization (e.g., as you suggested), or by spectral properties (as we did in this paper), and probably also in other ways. Regarding simplicity, we assume it depends on how we choose to define “simple” - is it by the cardinality of the frame? or maybe the computational complexity of computing the frame? We opted for the spectral method since it is relatively efficient to compute.
> We find these questions very interesting and believe they could lead to further developments in equivariant deep learning in future research.
>
>
>
> **Q:** Are the number of elements output by frames the smaller the better, or is there a balance between performance and computational efficiency?
> **A:** In terms of expressive power, the size of the frame does not matter. This can be seen by our expressive power theorem and corollaries. Another thing that is clear is that large frames are more computationally demanding than small frames. Regarding generalization performance, this is less obvious; we do believe different frame choices will inject different inductive bias and generalize differently. We leave this important and non-trivial question to future work.
>
>
> **Q:** How stable are the proposed frames? That is to say, if I add noise to the inputs, will the output subset of groups be significantly different?
> **A:** Thank you for this question. The stability of the frames depends of course on the particular definition of a frame, and as this reviewer noticed many constructions of equivariant frames are indeed possible. In this paper we mostly use spectral properties of matrices, in particular eigenvectors, to define frames. On the theoretical level we can analyze the stability of such frames for the generic case of simple spectrum (i.e., all eigenvalues have multiplicity one). The fact that this is indeed the generic case is justified in [1]. Next, stability results such as [2] (Theorem 8.1.12) can bound the change in frame as a function of the perturbation (i.e., noise) and the distance to the nearest eigenvalue. We added the relevant discussion in Section 3.1 (immediately after proposition 1). Furthermore, in Appendix B we added an empirical analysis of the stability of frames.  Our main observation is that the frames behave in a stable manner, i.e., small noise added to the input results in a small change of the frame. Please see the added experiment for more details.
>
>
> [1] On the geometry of the set of symmetric matrices with repeated eigenvalues, Breiding, Kozhasov, Lerario, Arnold Mathematical Journal 4(3), 2018, Springer.
>
> [2] Matrix computations, Gene H Golub, and Charles F Van Loan, 1996, Johns Hopkins University Press, Baltimore, MD.

---

> > ### Comment · Reviewer_57mq · 2021-11-22
> > **All my questions are answered, I keep my original score.**
> >
> > I would like to thank the authors for their response. All the raised questions are properly addressed in their rebuttal and I have no further concerns.
> >
> > This is great solid work, and I will keep my original rating of 8.

---

### Official Review · Reviewer_CToH · 2021-11-10

**Correctness:** 2
**Technical Novelty And Significance:** 4
**Empirical Novelty And Significance:** 3
**Recommendation:** 8
**Confidence:** 4

**Main Review:**

Strengths:
-	The paper proposes a very practical strategy of building equivariant nets
-	The universality proof helps convince the reader to use this method
-	The paper considers and experiments on three different instantiations of their method, showing wide applicability.
-	The experimental results show the method performs competitively.

Weaknesses:
-	I don’t understand what’s happening in theorem 4. It considers a subsample $\hat\mu$ of F(X) to be ‘good’ when the symmetrizer that uses the subsample is epsilon-close to the full F(X) symmetrizer. Then it says that the probability of *one particular* good subsample is bounded below.
However, that bound seems vacuus, as plugging any reasonable number brings the bound quickly close to 0. Also, it’s counter-intuitive why the bound should become looser as epsilon grows or as k grows.
What one would want instead is giving a lower bound of the probability that we get *any* $\hat\mu$ that is epsilon-close to the full F(X) symmetrizer. And we want this bound to get higher when epsilon or k increases.
The line below theorem 4 draws a conclusion that would follow from a theorem as I propose it above, not from the theorem in the paper.
As it is currently stated – and I’m not completely misunderstanding – theorem 4 can best be removed from the paper.
-	It is a bit unclear when the results apply to finite and infinite groups and frames F(X). Everywhere, a summation symbol is used, but in some places, F(X) is infinite. In the infinite cases, which measure should then be used? Can one always use some canonical Haar-like measure? In particular, in the proofs of theorem 1 and theorem 4 this should be discussed.
-	The writing of the paper can be improved. I don’t follow the choice of F(X) for the E(d) case. Which are the 2^d O(d) matrices? Perhaps the authors can elaborate in more detail one of the examples how to construct F(X) in the main paper, and then do the other two in the appendix.
-	I would like some more theoretical discussion about the choice of F(X). Does the choice of F(X) affect the output? If so, how? Is F(X) required to be continuous / does a continuous (non-trivial) F(X) always exist? What does the topology on 2^G look like? How does this affect the continuity of the symmetrized function? If it is discontinuous, does that affect the universality? When can F(X) be chosen to be finite?

Other comments:
-	Why does GA-MLP and GA-GIN+ID only get 50% score on EXP-classify? Are you there using a finite subsample of G or F(X)? And could you give any insight into why we’d expect then complete failure for a G subsample and complete success for a F(X) subsample?
-	In the proof of theorem 5, which norm is used for ||rho_2(g)|| ? It can’t be the max K-norm because K is a subset of the input of phi, not the output. Is it the operator norm?
-	I would like theorem 1 to be put in the main paper, as it shows why the key construction is correct.
-	I think a citation would be appropriate to Finzi et al 2020, “Generalizing Convolutional Neural Networks for Equivariance to Lie Groups on Arbitrary Continuous Data”, as they also consider sampling from the group to build equivariant networks.
-	Why have the authors chosen the name “frame” for F(X)? I know frame as a set of vectors or in the context of a frame bundle.


**Summary Of The Paper:**

The paper proposes to make any neural network equivariant by symmetrizing over a subset of the group, rather than over whole group. If the subset selection F(X), depending on input X, is equivariant (gFX=FgX), then the symmetrization is equivariant.
The authors furthermore prove:
1)	When interested in invariant prediction, the subset can be chosen in the quotient G/G_X, where G_X is the stabilizer subgroup of X.
2)	When symmetrizing with a random subsample of F(X), the probability of a particular subsample that deviates from symmetrizing with all of F(X) by less than some epsilon, is bounded below.
3)	When using the symmetrization of a universal model, the resulting model class is universal in the class of equivariant functions.


**Summary Of The Review:**

I think this is a great paper, as it proposes a new practical method for building equivariant networks which is broadly applicable, universal and performs well in practice. I have serious concerns about theorem 4. If the authors convince me why it makes sense, or if they remove it, I will increase my score.

I've updated my score after the response and revision.

---

> ### Author Response · Authors · 2021-11-19
> **Official Response to Reviewer CToH  1/2**
>
>
> Thank you for your detailed review. Below we address the main comments and questions expressed in this review. All changes refer to the uploaded revision.
>
> **Q:** Theorem 4 is unintuitive (behavior of $\epsilon,k$) and unclear (bounds probability of
> one subsample at a time), please remove from paper.
> **A:** The idea behind Theorem 4 was to compare the lower bound for a **fixed** $k$ and $\epsilon$, which in this case show that smaller $m_F$ leads to a better lower bound. The counter-intuitive behavior of this bound w.r.t.~$k$ and $\epsilon$ stems from the fact that the size of the set of "good" empirical distributions $\hat{\mu}$ is increasing with $k$ and $\epsilon$. Nevertheless, we agree with the reviewer that this theorem does not provide the full picture and moved it to supplementary, adding the above disclaimers. We do feel it can be used in the future as a step to provide a full proof of approximation quality of the FA approximator, and therefore believe it's a good idea to keep it in the supplementary.
>
>
> **Q:** “It is a bit unclear when the results apply to finite and infinite groups and frames $F(X)$. Everywhere, a summation symbol is used, but in some places, $F(X)$ is infinite. In the infinite cases, which measure should then be used? Can one always use some canonical Haar-like measure? In particular, in the proofs of theorem 1 and theorem 4 this should be discussed.”
> **A:** In this paper we only treat the finite frame case. We added a clear statement at Section 2.1 that in this paper we only consider $X, F$ so that $F(X)$ are finite sets. We also added related comments in other relevant places in the paper: In Section 2.2 dealing with expressive power, we explicitly state we work in domains where the frame is finite. The definition of frames in 3.1 are defined only for simple spectrum covariance matrices. In Corollary 2 we also added the finite frame assumption.
> Regarding the general (infinite) case we believe this is an important future direction. Yes, Haar measure can be used but it won’t cover all interesting cases (e.g., think of 1-dimensional rotations embedded in 3-dimensional rotations). The most general treatment seems to require data dependent measures, but this is out of scope for this paper and will be treated in a future publication.
>
>
> **Q:** “I don’t follow the choice of $F(X)$ for the $E(d)$ case. Which are the $2^d$ $O(d)$ matrices?”
> **A:** We rewrote this part in the paper (see Section 3.1), hopefully it is clearer.
>
> **Q:** I would like some more discussion about the choice of $F(X)$. Does the choice of $F(X)$ affect the output? If so, how? Is $F(X)$ required to be continuous / does a continuous (non-trivial) $F(X)$ always exist? How does this affect the continuity of the symmetrized function? If it is discontinuous, does that affect the universality?
> **A:** The choice of frame affects the output: the frame sets a “normalized form” of the input to be injected into the network. Therefore different frame choices inject different inductive biases.  Regarding continuity: in the continuous group case (i.e., $E(d)$) the frames are continuous where they are defined, see discussion in Section 3.1, after Proposition 1, and Appendix B. In the discrete group setting (e.g., $S_n$) we are not sure what continuity means. Globally continuous frames are not always defined. A similar situation is in differential geometry where choosing continuous frame fields can fail due to topological obstructions or geometrical (e.g., umbilical points). Universality is not affected by discontinuous frames; we have simplified and generalized the universality claim and proof, and as can be checked, it does not require continuity of the frame, see Section 2.2.
>
>
> **Q:** Why does GA-MLP and GA-GIN+ID only get 50% score on EXP-classify? Are you there using a finite subsample of G or $F(X)$? And could you give any insight into why we’d expect then complete failure for a G subsample and complete success for a $F(X)$ subsample?
> **A:** The core reason for the difference between the FA and GA sampling method lies in Figure 2 (left) in section 5.2. The plot describes how well each sample method is able to approximate its average as a function of the number of samples. As can be seen from the plot, when $k=1$ (which is the same number of samples used in the EXP experiment) the variance of the frame sampling is smaller than the variance of the group sampling. The difference in variance explains the accuracy gap in the EXP task. We can expect the same phenomenon with GIN+ID.

---

> > ### Author Response · Authors · 2021-11-19
> > **Official Response to Reviewer CToH 2/2**
> >
> > **Q:** In the proof of theorem 5, which norm is used for $||\rho_2(g)||$ ? It can’t be the max K-norm because K is a subset of the input of phi, not the output. Is it the operator norm?
> > **A:** It is the operator norm, we clarified this and the other norms using specific notation (see Definition 1 and Appendix A.5).
> >
> >
> > **Q:** I think a citation would be appropriate to Finzi et al 2020, “Generalizing Convolutional Neural Networks for Equivariance to Lie Groups on Arbitrary Continuous Data”, as they also consider sampling from the group to build equivariant networks.
> > **A:** citation was added in the related work section.
> >
> > **Q:** Why have the authors chosen the name “frame” for $F(X)$? I know frame as a set of
> > vectors or in the context of a frame bundle.
> > **A:** As shortly alluded to above, our intuition comes from differential geometry where the
> > method of moving frames, and in particular Darboux frames, are chosen in an $E(3)$ equivariant manner by aligning them with principal directions and the normal to the surface. Further note that also Darboux frames are defined everywhere except at umbilical points, which are defined by repeating eigenvalues of the shape operator, similar to the situation with our $E(d)$ frame construction.

---

> > > ### Comment · Reviewer_CToH · 2021-11-19
> > > **Very satisfied**
> > >
> > > Thanks for your response and extensive revisions. My questions have been excellently answered and I'll adjust my score.

---

### Decision · Program_Chairs · 2022-01-20

**Decision:**

Accept (Oral)

**Comment:**

The submission proposes a method to make a pre-existing model equivariant to desired symmetries: frame averaging. The strategy relies on a significant reduction of the number of symmetries to average over (with respect to the Reynolds operator) and uniform subsampling. The paper also demonstrates the usefulness of this method theoretically (universal approximation result) and practically (competitive performance). The contributions are clear and the core idea is simple.
I recommend this paper for acceptance with spotlight.